# A model for network-based identification and pharmacological targeting of aberrant, replication-permissive transcriptional programs induced by viral infection

Pasquale Laise[1,2,19], Megan L. Stanifer[3,4,19], Gideon Bosker[1], Xiaoyun Sun[1], Sergio Triana[5,6], Patricio Doldan[7,8], Federico La Manna[9,10,11,12], Marta De Menna[9,10,11,12], Ronald B. Realubit[2], Sergey Pampou[2], Charles Karan[2], Theodore Alexandrov[5,13,14], Marianna Kruithof-de Julio [9,10,11,12], Andrea Califano [2,15,16,17,18✉], Steeve Boulant[4,7,8✉] & Mariano J. Alvarez [1,2✉]

SARS-CoV-2 hijacks the host cell transcriptional machinery to induce a phenotypic state amenable to its replication. Here we show that analysis of Master Regulator proteins representing mechanistic determinants of the gene expression signature induced by SARS-CoV-2 in infected cells revealed coordinated inactivation of Master Regulators enriched in physical interactions with SARS-CoV-2 proteins, suggesting their mechanistic role in maintaining a host cell state refractory to virus replication. To test their functional relevance, we measured SARS-CoV-2 replication in epithelial cells treated with drugs predicted to activate the entire repertoire of repressed Master Regulators, based on their experimentally elucidated, context-specific mechanism of action. Overall, 15 of the 18 drugs predicted to be effective by this methodology induced significant reduction of SARS-CoV-2 replication, without affecting cell viability. This model for host-directed pharmacological therapy is fully generalizable and can be deployed to identify drugs targeting host cell-based Master Regulator signatures induced by virtually any pathogen.

[1] DarwinHealth Inc, New York, NY, USA. [2] Department of Systems Biology, Columbia University Irving Medical Center, New York, NY, USA. [3] Department of Infectious Diseases, Molecular Virology, Heidelberg University Hospital, Heidelberg, Germany. [4] Department of Molecular Genetics and Microbiology, University of Florida, College of Medicine, Gainesville, FL, USA. [5] Structural and Computational Biology Unit, European Molecular Biology Laboratory, Heidelberg, Germany. [6] Collaboration for joint PhD degree between EMBL and Heidelberg University, Faculty of Biosciences, Heidelberg, Germany. [7] Department of Infectious Diseases, Virology, Heidelberg University Hospital, Heidelberg, Germany. [8] Research Group "Cellular Polarity and Viral Infection", German Cancer Research Center (DKFZ), Heidelberg, Germany. [9] Department for BioMedical Research, Urology Research Laboratory, University of Bern, Bern, Switzerland. [10] Translational Organoid Resource, Department for BioMedical Research, University of Bern, Bern, Switzerland. [11] Bern Center for Precision Medicine, University of Bern and Inselspital, Bern, Switzerland. [12] Department of Urology, Inselspital, Bern University Hospital, Bern, Switzerland. [13] Skaggs School of Pharmacy and Pharmaceutical Sciences, University of California San Diego, La Jolla, CA, USA. [14] Molecular Medicine Partnership Unit (MMPU), European Molecular Biology Laboratory, Heidelberg, Germany. [15] Herbert Irving Comprehensive Cancer Center, Columbia University Irving Medical Center, New York, NY, USA. [16] Department of Medicine, Columbia University Irving Medical Center, New York, NY, USA. [17] Department of Biochemistry & Molecular Biophysics, Columbia University Irving Medical Center, New York, NY, USA. [18] Department of Biomedical Informatics, Columbia University Irving Medical Center, New York, NY, USA. [19]These authors contributed equally: Pasquale Laise, Megan L. Stanifer. ✉email: ac2248@cumc.columbia.edu; s.boulant@ufl.edu; malvarez@darwinhealth.com

Several approaches have been employed to identify specific host cell pathways and proteins whose individual interaction with viral proteins is either required to mediate SARS-CoV-2 infection or that represents key modulators of virulence[1–6]. In contrast, a paucity of effort has been devoted to elucidating the host cell transcriptional control mechanisms and programs hijacked by viruses, including identification of the Master Regulator (MR) proteins that effect infection-mediated reprogramming of the host cell transcriptional state. MRs are proteins whose activity is necessary and sufficient to maintain the transcriptional identity of a specific cellular phenotype. They are organized in highly inter-regulated protein modules, or transcriptional regulatory checkpoints, which operate as a molecular switch, controlling the transcriptional identity of both physiologic and pathologic cell states (for a recent perspective, please see ref. [7]). More importantly, there has been no experimental evaluation of the role of such host MR proteins in the virus life cycle nor their amenability to pharmacological targeting for the purpose of inhibiting viral replication.

Here, we show that specific host MR proteins, representing viral infection-mediated determinants of the transcriptional regulatory programs hijacked by viruses, are required for establishing a host-cell phenotypic state amenable to virus replication. Specifically, we leveraged an established systems biology-based methodology, originally developed in the field of oncology[7], to identify MR proteins that mechanistically control the transcriptional state of virus-infected cells. We then prioritized drugs capable of inverting the activity of MR proteins—thus decommissioning the regulatory programs induced by viral infection to maintain a pro-infective cell state—using another oncology-based approach described in ref. [8]. We propose that extension and translation of these cancer-based methodologies to study viral infection can identify host cell MR proteins representing key mechanistic determinants of virus-mediated host cell reprogramming, as well as the drugs that can abrogate this transition.

As we have previously shown, MRs can be accurately and systematically identified by assessing the enrichment of their transcriptional targets in differentially expressed genes, using the Virtual Inference of Protein activity by Enriched Regulon analysis (VIPER)[9]. While many approaches can be used to identify the tissue-specific targets of a regulatory protein, the Algorithm for the Accurate Reconstruction of Cellular Networks (ARACNe)[10] is among the few that have been extensively experimentally tested, with validation rates exceeding 70%[10–12]. We have shown that VIPER can accurately measure the activity levels of >70% of regulatory proteins, including in single cells, where we have shown that metaVIPER[13]—a VIPER extension specifically designed for single-cell analyses—can virtually eliminate the gene dropout issue due to low single cell profiling depth;[14,15] and, notably, outperforms antibody-based measurements[14]. Hereafter, for simplicity, we will refer to the transcriptional activity inferred by VIPER or metaVIPER, as protein activity. The combination of these two algorithms has been highly effective in elucidating protein-based mechanisms that were virtually undetectable by gene expression-based methods alone[7,14,16,17] (see methods for additional details). Moreover, once MR protein activity levels are quantified by VIPER analysis, the CLIA-certified OncoTreat algorithm[8] can accurately and efficiently identify small molecule inhibitors that can invert their activity (MR-inverter drugs), thereby abrogating the regulatory programs they control. The OncoTreat algorithm leverages large-scale gene expression profiles of MR-matched cell lines perturbed with a comprehensive repertoire of clinically relevant drugs, including Food and Drug Administration (FDA)-approved and late-stage experimental agents, and has led to several clinical trials evaluating drug therapy for cancer (NCT02066532, NCT02632071, and NCT03211988, among others).

Given the urgency and unmet needs mandated by the COVID-19 pandemic, we proceeded to test the applicability of this model to SARS-CoV-2 infection. Specifically, we asked whether this methodology could be used to identify host cell MR proteins representing the mechanistic determinants of the transcriptional programs hijacked by the virus to support efficient replication and, by extension, whether we can identify drugs capable of inverting their activity, thereby making host cells more resistant to hijacking and viral replication. The methodology can be trivially generalized to other pathogens, conditional only on the availability of appropriate infection gene expression signatures.

VIPER-inferred MRs from multiple SARS-CoV-2 infection models consistently showed that the host MR proteins that were *significantly activated* following SARS-CoV-2 infection controlled innate immune response programs. This suggests that the transcriptional programs supporting optimal viral replication and infectivity, during the hijack phase, may be controlled by host MRs that were *significantly inactivated* following infection. Supporting this hypothesis, we found the inactivated MRs to be highly enriched in interactions with SARS-CoV-2 proteins and in genes reported as essential antiviral factors by CRISPR screens[2,4,6]. To further test this hypothesis, we adapted the OncoTreat algorithm[8] to prioritize compounds based on their ability to activate the entire set of virus-inactivated MR proteins, and evaluated their effect on SARS-CoV-2 replication in infected epithelial cell cultures. Prioritization of 154 FDA-approved drugs —primarily for use in oncology—was highly effective, with 15 out of 18 predicted drugs effectively reducing SARS-CoV-2 replication in colon epithelial cells, with no significant reduction of cell viability. Based on these findings, we conclude that SARS-CoV-2-induced transition of the host cell phenotypic state is required for its optimal replication. Moreover, we provide a model for systematically dissecting the MR proteins that mechanistically facilitate this transition and for identifying MR-inverting drugs that, by blocking this phenotypic transition, can induce a host cell regulatory state of "viral contraception". This model, which we call, "ViroTreat", could be used to identify therapeutic options in the COVID-19 setting and can be easily generalized to virtually any viral pathogen-mediated host cell hijacking that is essential for the infective cycle.

## Results

**SARS-CoV-2-induced MR signature**. To elucidate the MR proteins mediating SARS-CoV2-induced host-cell phenotypic transition, we analyzed publicly available single cell (scRNASeq) profiles of SARS-CoV-2 infected epithelial cells (Supplementary Table 1), including epithelial cell lines from both lung adeno-carcinoma (Calu-3 and H1299)[18], and gastrointestinal organoid models from the ileum and colon[19]. Single cell RNASeq analysis allows highly effective identification of individual virus-infected cells, which would otherwise represent only a minority of cells in culture. Moreover, single cell-based gene expression signatures—computed by comparing confirmed infected cells to non-infected controls—are less affected by contamination and dilution effects typical of bulk RNASeq signatures representing a mixture of infected and non-infected cells (Supplementary Fig. 1 and Methods).

The differential activity of 5,734 proteins, including 1,723 transcription factors, 630 co-transcription factors, and 3,381 signaling proteins, was estimated for each infected cell with the VIPER algorithm[9] (Supplementary Data 1). This analysis revealed highly conserved differential protein activity signatures, as defined by the conservation of top 50 most differentially active candidate MRs. The use of 50 proteins is based on previous results in the context of cancer, where less than 50 candidate MRs

are required to canalize the effect of each individual patient genetics on the cell transcriptional identity[16]. By analogy to tumor MRs[7], we will refer to this repertoire of virus-induced MRs as the Viral CheckPoint. The analysis identified a highly conserved MR core induced by SARS-CoV-2 infection, within each available cellular model, across all post-infection time-points for which data was available ($p < 10^{-40}$, by 2-tailed aREA test[9], Fig. 1a and Supplementary Fig. 2a).

When comparing equivalent time-points, we observed significant conservation of the differentially active protein signature across lineage-related cell models (e.g., Calu-3 vs. H1299, at 12 h, $p < 10^{-40}$, Supplementary Fig. 2a). Interestingly, the virus-mediated MR signature was highly conserved even across unrelated lineages, when equivalent time-points were considered

(e.g., H1299 vs. colon non-transformed organoid at 24 h, $p < 0.01$, Supplementary Fig. 2a). Taken together, these findings suggest the existence of a highly reproducible, SARS-CoV-2-mediated MR activity signature in epithelial cells, regardless of organ context (lung vs. gastrointestinal (GI)). Interestingly, however, *inactivated* MRs were significantly more conserved than *activated* MRs, both across models and lineages ($p < 10^{-6}$, 2-tailed paired U-test, Supplementary Fig. 2b, c), suggesting a potentially distinct biological role for the activated vs. inactivated components of the SARS-CoV-2 MR core.

The MR activity signatures detected by single cell analyses were also recapitulated by bulk-tissue analysis of SARS-CoV-2-infected epithelial cells (ST1), albeit at a slightly lower statistical significance, as we expected. These findings applied to bulk-tissue analysis of

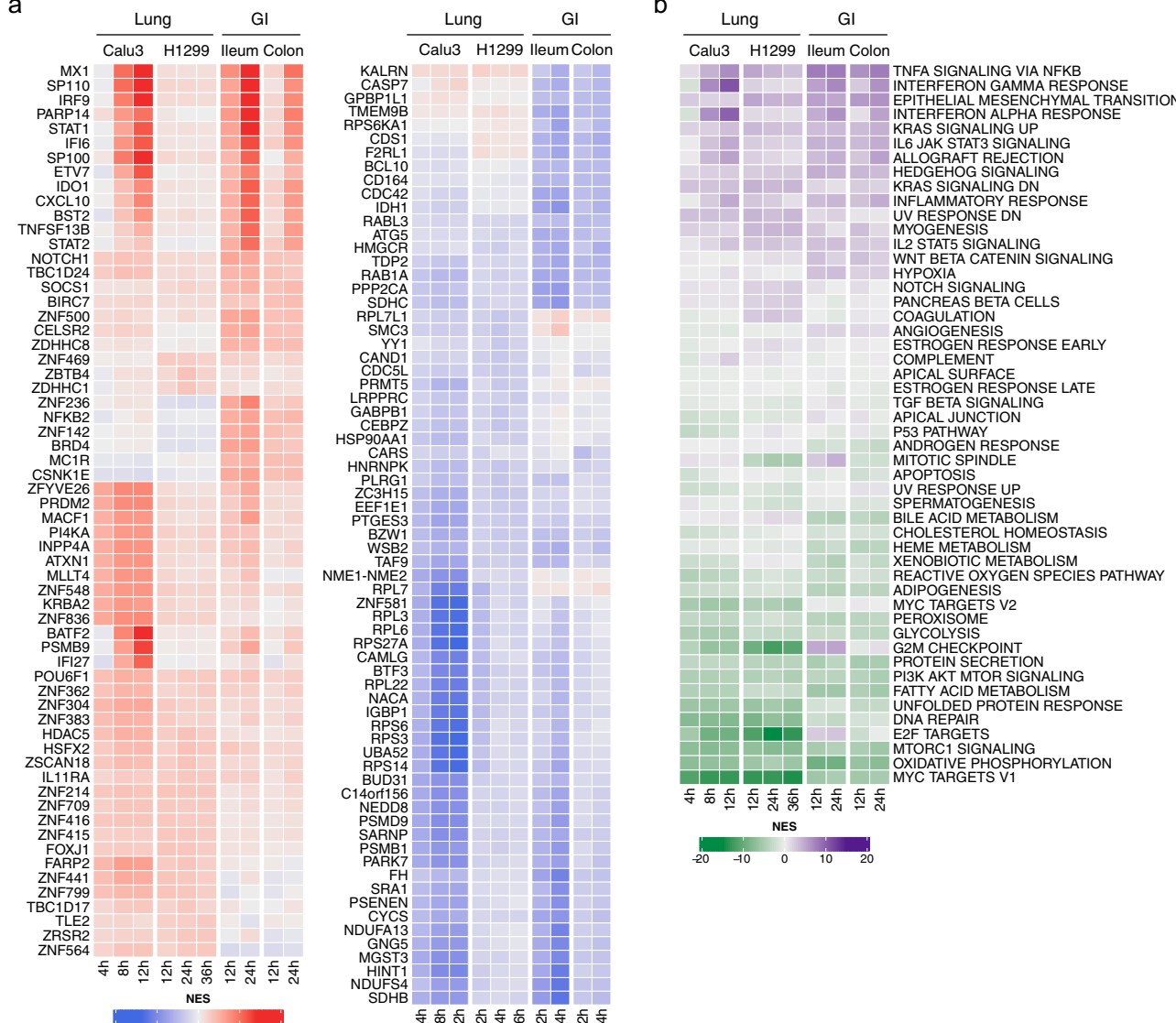

**Fig. 1 Changes in host cell protein activity in response to SARS-CoV-2 virus infection. a** Left, heatmap showing the VIPER-inferred differential activity of the top 10 most activated proteins in response to SARS-CoV-2 infection in each of the models and time-points profiled (62 proteins across all evaluated conditions) at the single-cell level. Right, heatmap showing the activity of the top 10 most inactivated proteins in response to SARS-CoV-2 infection in each of the models and time-points profiled (69 proteins across all evaluated conditions) at the single-cell level. Differential protein activity is expressed in Normalized Enrichment Score (NES) units with protein inactivation and activation induced by SARS-CoV-2 infection shown in blue and red color, respectively. **b** Heatmap showing the enrichment of biological hallmarks in the SARS-CoV-2-induced protein activity signatures. Shown is the NES estimated by the aREA algorithm, with purple color indicating enrichment in the over activated proteins and green color indicating enrichment in the inactivated proteins.

both transformed models, including lung (Calu-3, H1299, and A549) and colon (Caco-2) adenocarcinoma, and normal human bronchial epithelial (NHBE) primary cells, as well as to more physiologic models, including lung organoids. As should be expected, MR conservation was more significant for models characterized by high infection rates (Supplementary Fig. 2a), likely due to signature dilution/contamination by a high proportion of non-infected cells in other models.

**MRs govern distinct biological functions**. Gene Set Enrichment Analysis (GSEA)[20] demonstrated a critical dichotomy of biological hallmark programs enriched in activated vs. inactivated MRs (Fig. 1b). Specifically, biological hallmarks enriched in *activated* MRs included inflammatory response, epithelial-to-mesenchymal transition (EMT) and interferon response. Indeed, among the top aberrantly activated MRs, we identified MX1, a protein induced by interferon I and II[21], the interferon regulator IRF9, and additional transcriptional regulators that mediate cellular response to interferons, such as STAT1 and STAT2[22] (Fig. 1a).

In contrast, our model shows that biological hallmarks enriched in *inactivated* MRs were strongly related to virus-mediated host-cell hijacking programs, such as PI3K signaling, unfolded protein response, DNA repair, and metabolic-related processes[23,24] (Fig. 1b). Consistent with this observation, the most significantly inactivated MRs included several ribosomal subunit members (such as RPS27A, RPS3, RPL3, RPS6, RPS14), as well as proteins involved in cell cycle arrest (UBA52)[25], translational regulation, and cellular metabolism (GABPB1)[26] (Fig. 1a).

**VIPER-inferred MRs are enriched in SARS-CoV-2-interacting proteins**. To assess whether activated vs. inactivated MRs in our model may represent a more effective target for drug-mediated reversal, we proceeded to assess whether either class was enriched in host proteins previously identified as cognate binding partners of SARS-CoV-2 proteins. For this analysis, we leveraged a collection of 332 host proteins previously reported to be involved in protein-protein interactions (PPIs) with 26 of the 29 proteins encoded by the SARS-CoV-2 genome, as determined by mass-spec analysis of pull-down assays[2]. Of these interactions, 90 were with proteins included in the 5,734 we analyzed by VIPER. GSEA[20] revealed statistically significant enrichment of these 90 proteins in SARS-CoV-2 *inactivated* but not *activated* MRs, across all the evaluated single-cell protein activity signatures ($p < 10^{-3}$, 2-tailed GSEA, Supplementary Fig. 3). This suggests that host cell proteins that physically interact with SARS-CoV-2 proteins are mostly inactivated in response to the infection.

**VIPER-inferred MRs are enriched in viral infection-essential genes**. To further confirm the functional duality of the inferred MRs, we also assessed their enrichment in genes previously reported as essential to the virus infectious cycle. Specifically, we evaluated their enrichment in genes identified by functional CRISPR screens from two different studies, including using SARS-CoV-2 infected Vero[6] and Huh-7.5[4] cells. Consistent with our original observation and definition of the SARS-CoV-2 induced MR signature, the 50 most inactivated candidate MRs—as determined by integrating results from both lung and GI models—were significantly enriched in infection-essential genes identified in both CRISPR screen ($p < 10^{-4}$ and $p < 10^{-3}$, respectively), as well as in the integrated set (Supplementary Fig. 4a–c, $p < 10^{-4}$). In contrast, the 50 most activated MRs were not significantly enriched in infection essential genes (Supplementary Fig. 4d–f). Confirming these results, analysis of two additional CRISPR screen reports showed enrichment of the 50 most inactivated SARS-CoV-2 infection candidate MRs on

infection-essential genes identified in A549-ACE2 cells[3], as well as significant enrichment of the infection-essential genes identified in Calu-3 and Caco-2 cells[27] among the candidate MR proteins being inactivated in response to SARS-CoV-2 infection (Supplementary Fig. 4g–i).

**ViroTreat prioritization of FDA-approved drugs**. We have previously shown that tumor checkpoints can be pharmacologically switched, either off[8,12,17,28,29] or on[16], leading to their collapse and loss of viability or gain of associated functional properties, respectively. This observation was instrumental for the development and validation of the New York Clinical Laboratory Improvement Amendments (CLIA) certified, VIPER-based methodology OncoTreat, for the prioritization of small molecule compounds that can either inactivate or activate a tumor checkpoint on a sample-by-sample basis, with critical applications in precision oncology[8]. To test the dependence of SARS-CoV-2 replication on inactivation of the MR proteins—termed Viral Checkpoint for analogy to tumor Checkpoints[7]—we adapted the OncoTreat algorithm[8] to identify small molecule compounds capable of activating such MRs (ViroTreat, Fig. 2). We hypothesize that such drug-induced effects would keep the host cell phenotype in a "viral contraception" regulatory state that effectively reduces the viral replication rate.

We have shown that drug Mechanism of Action (MoA)—as represented by the proteins that are differentially activated/inactivated—is an effective predictor of drug activity in vivo and in explants[30,31]. This is assessed by VIPER analysis of MR-matched cell lines following perturbation with a large repertoire of drugs, at the highest sublethal concentration ($IC_{20}$), as assessed by dose response curves. The PanACEA database (PANcancer Analysis of Chemical Entity Activity)[32] comprises drug perturbation RNA-seq profiles representing 25 cell lines and an average of 350 drugs per cell line. Among these, the LoVo and NCI-H1973 cell lines were identified as those whose lineage matched the GI epithelial and lung epithelial cell models used for SARS-CoV-2 infection assays, respectively. However, while LoVo (human colon cell line) showed statistically significant MR protein conservation ($p < 10^{-5}$ by OncoMatch analysis[30]), when compared with the colon adenocarcinoma cell line susceptible to SARS-CoV-2 infection (Caco-2[33], Supplementary Fig. 5a, b), such conservation was not observed between NCI-H1793 cells and any of the three lung cell lines susceptible to SARS-CoV-2 infection (Calu-3, ACE2-A549 and H1299, Supplementary Fig. 5c–h). We have previously shown that recapitulation of MR protein activity by the drug perturbed models is important to maximize drug MoA conservation and OncoTreat analysis sensitivity[30,34]. Based on these results and considering availability of a compatible cell line as a relevant validation model to experimentally assess ViroTreat-predicted drugs, for this model we focused our validation efforts on the GI context.

VIPER was used to elucidate the MoA of 154 FDA-approved oncology drugs, where MoA is defined as the repertoire of proteins differentially activated/inactivated at 24 h following drug perturbation. While this was done specifically in colon epithelial cells for this study, the analysis can be easily extended to assess drug MoA in other cellular contexts. Specifically, the RNA-seq profiles used in this analysis were generated at 24 h (by PLATE-Seq assays[35]), following treatment of a colon adenocarcinoma cell line (LoVo) with a library of FDA-approved drugs and vehicle control (DMSO). To avoid assessing cell death or stress mechanisms, rather than drug MoA effects, drugs were titrated at their highest sublethal concentration (i.e., their 48 h $IC_{20}$), as assessed by 10-point dose response curves (see methods for additional details). Resulting profiles were then used to assess the

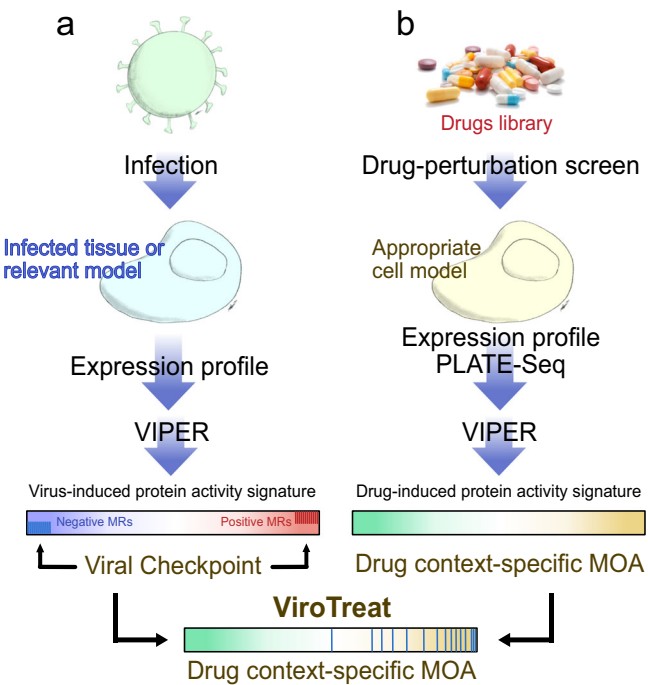

**Fig. 2 Schematic representation of the ViroTreat algorithm. a** Virus-induced MR proteins—the Viral Checkpoint—dissected by VIPER analysis of a gene expression signature, obtained by comparing an infected tissue or relevant model with non-infected mock controls. **b** Context-specific drug MoA database, generated by perturbing an appropriate cell model with therapeutically relevant drug concentrations, followed by VIPER analysis of the drug-induced gene expression signatures to infer the drug-induced protein activity signature. ViroTreat prioritizes drugs able to activate the Viral Checkpoint's negative MR proteins by quantifying the enrichment of such proteins on the drugs' context-specific MoA.

differential activity of regulatory proteins in drug vs. vehicle control-treated cells with the VIPER algorithm[9]. Finally, drugs were prioritized based on their ability to activate the MR proteins inactivated by SARS-CoV-2 infection, as assessed by their enrichment in proteins differentially activated by each drug, using the aREA algorithm[8,9] (Fig. 2). Similar to GSEA[20], aREA estimates the statistical significance for the enrichment of a set of genes or proteins on a vector of differentially expressed genes or differentially active proteins[9].

ViroTreat predictions were averaged across available GI organoid models and across all evaluated time points. Among the 154 FDA-approved drugs profiled in LoVo cells, ViroTreat prioritized 22 (13 orally available and 9 intravenous) at a highly conservative statistical threshold ($p < 10^{-5}$, Bonferroni corrected (BC)), see Fig. 3 and Supplementary Data 2), based on their predicted activating effect for the proteins that showed the strongest inactivation in response to SARS-CoV-2 infection.

**ViroTreat-predicted drugs inhibit SARS-CoV-2 replication.** To provide proof-of-concept validation for the ViroTreat predictions in our model, we first assessed drug-mediated inhibition of SARS-CoV-2 replication by ViroTreat-predicted vs. control drugs in the colon adenocarcinoma cell line (Caco-2) known to support SARS-CoV-2 infection[33].

For this assay, we considered all 13 ViroTreat-inferred orally-available drugs, as a more clinically relevant group, and the top 5 most significant intravenous (IV) drugs. As candidate negative controls, we selected 12 drugs—including 8 orally available agents and 4 IV drugs—not inferred as statistically significant by

ViroTreat ($p \geq 0.01$, Fig. 3 and Supplementary Data 2). Caco-2 cells were pre-treated for 24 h prior to SARS-CoV-2 infection. Drug concentration was maintained through the entire infection time course and the relative virus replication levels and cell viability were assessed by immunofluorescence staining 24 h post-infection (see methods and Fig. 4a). This approach allows us to address the impact of the individual drugs on the number of SARS-CoV-2 infected cells but does not inform whether restriction takes place at the viral entry level or during viral genome replication. For each drug, the viability-normalized effect on SARS-CoV-2 replication (antiviral effect) was quantified as the log-ratio between viral replication and cell viability reduction relative to vehicle-treated (DMSO) controls (Supplementary Fig. 6). Since multiple concentrations were tested, the lowest concentration corresponding to a significant antiviral effect with no or limited cytotoxic effect to the cells was reported (Supplementary Data 2). As a proof-of-concept for the ability of this model to identify drugs capable of reducing replication of SARS-CoV-2, we considered drugs to be validated only if their antiviral effect was statistically significant (FDR < 0.05) and they induced a decrease in virus replication of at least 20%. This additional condition was used to further increase the stringency when considering the antiviral effect of a drug (see Methods).

Of 18 drugs predicted to activate the MR proteins inactivated by SARS-CoV-2 infection, 15 (83%) showed statistically significant antiviral effect. In contrast, none of the 12 drugs selected as potential negative controls showed significant antiviral effect (Fig. 4b and Supplementary Data 2), demonstrating a significant enrichment of ViroTreat results in drugs with antiviral activity ($p < 10^{-5}$, 1-tailed Fisher's exact test (FET)). Consistently, the Receiver Operating Characteristic (ROC) had an Area Under the Curve AUC = 0.907 (95% Confidence Interval: 0.77–0.91), which is highly statistically significant ($p < 10^{-4}$, Fig. 4c), demonstrating the predictive power of ViroTreat in this proof-of-concept.

To further assess the pathogen-specific nature of ViroTreat predictions, we tested the ability of the 8 ViroTreat-inferred drugs showing the strongest inhibition of SARS-CoV-2 replication, to inhibit rotavirus replication in Caco-2 cells. Interestingly, none of these drugs significantly impaired rotavirus replication (Supplementary Fig. 7 and Supplementary Data 2), showing that ViroTreat-inferred antiviral effects cannot be attributed to generalized impairment of host cellular functions universally required for viral replication, but rather to activation of host-cell MRs required for the maintenance of a host-cell phenotypic state specifically refractory to SARS-CoV-2 replication.

To also assess whether the antiviral activity of ViroTreat-predicted oncology drugs in Caco-2 cells might possibly be attributed to their antineoplastic effects in a cancer cell context, we evaluated the antiviral properties of the top 8 drugs in non-transformed, human GI organoid-derived 2D primary cell cultures. When tested in this more physiologic context, 7 of the 8 assayed drugs, including idarubicin, bosutinib, cyclosporine, bicalutamide, vorinostat, amiodarone and osimertinib, demonstrated significant antiviral effect against SARS-CoV-2 based on our original criteria (FDR < 0.05 and decrease in SARS-CoV-2 replication of at least 20%, Fig. 4d). Except for bicalutamide, which exerted its antiviral effect at a 125-fold higher concentration, all drugs were tested at concentrations comparable to their 48 h $IC_{20}$ in LoVo cells, representing the highest sub-toxic concentration usable for optimal MoA elucidation. These findings suggest that ViroTreat can apply the molecular characterization of a drug's MoA, as obtained by the measured effect of the drug on protein activity levels in tissue lineage-matched, neoplastic cell line models, to prioritize and repurpose drugs with potential antiviral activity in both infected tumor models as well as non-transformed human organoid-derived 2D primary cell cultures.

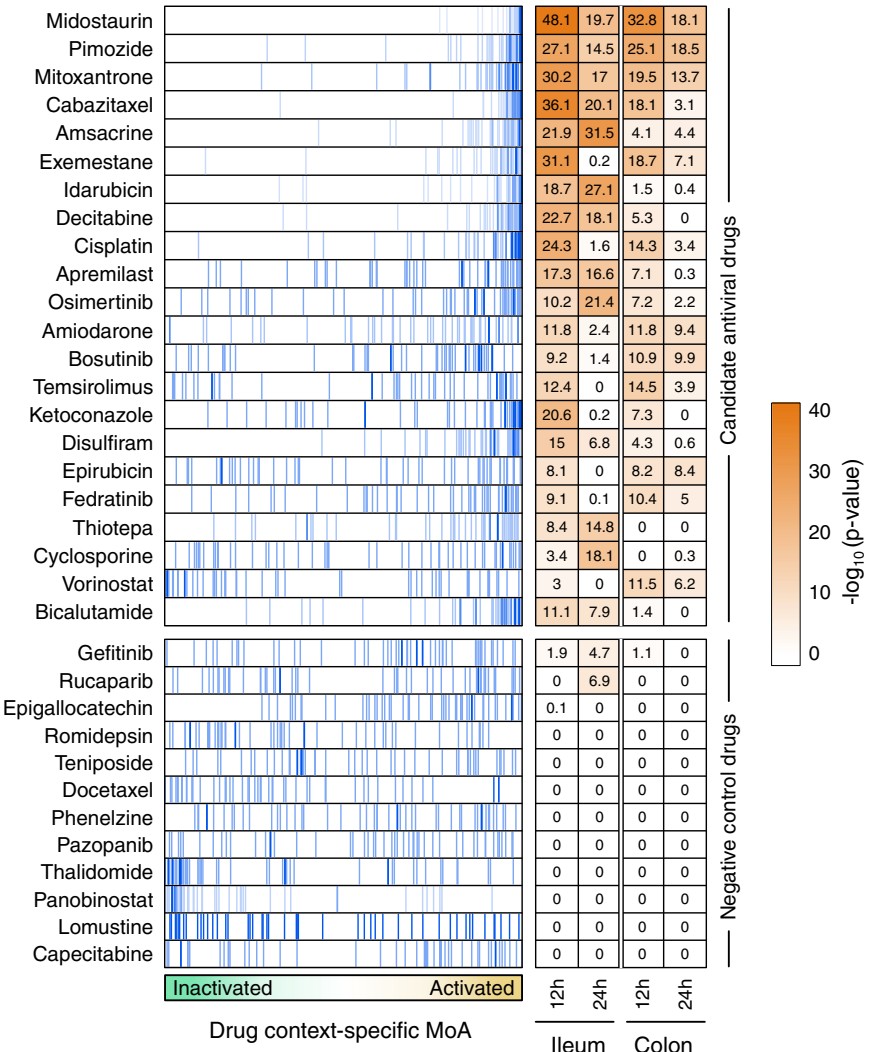

**Fig. 3 ViroTreat results for the GI models.** Shown are the enrichment plots for the top 50 most inactivated proteins (blue vertical lines), in response to SARS-CoV-2 infection (the negative component of the viral Checkpoint) of the ileum organoid for 12 h, on the protein activity signature induced by the drug perturbations—drug context-specific MoA, represented by the green-orange color scale in the *x*-axis—of LoVo colon adenocarcinoma cells. The heatmap shows the Bonferroni's corrected -log$_{10}$(*p*-value) estimated by ViroTreat. Shown are all the 22 candidate drugs (ViroTreat *p* < 10$^{-5}$) and 12 drugs selected as negative controls (ViroTreat *p* > 0.01) in both ileum and colon-derived organoids at 12 h and 24 h post-infection.

Finally, to test the tissue lineage context-specificity of ViroTreat predictions, we assessed the antiviral effect of the 8 ViroTreat predicted drugs for the GI context showing the strongest inhibition of SARS-CoV-2 infection in Caco-2, in lung adenocarcinoma cell line models (Calu-3 and ACE2-A549). Interestingly, only cyclosporine and osimertinib showed a significant antiviral effect (FDR < 0.05 and ≥20% virus replication decrease), while amiodarone, apremilast, bicalutamide, bosutinib, exemestane, and pimozide did not (Supplementary Fig. 8 and Supplementary Data 2). These results highlight the relevance of lineage context-specificity when prioritizing drugs with ViroTreat.

**Discussion**

We report here a model characterizing the regulatory biology of virus-host interaction, in which viral infection induces a phenotypic transition in the host cell toward a state that is promotive of viral replication. We applied Master Regulator (MR) inference analysis[9,16] to systematically dissect the transcriptional regulators (MR proteins) hijacked by the virus (Viral CheckPoint) and demonstrated, using a model of SARS-CoV-2 infection in gastrointestinal epithelial cells, that pharmacologically blocking this

transition is sufficient to maintain the host cell in a state of "transcriptional contraception" that is adverse to virus replication. We adapted the OncoTreat framework, originally developed to prioritize drugs for precision oncology[8], to identify drugs with concerted activity on the Viral Checkpoint.

We propose that the approach employed in this model, which we call ViroTreat, can be used as a mechanism-based framework for repurposing drugs, based on their ability to reprogram host cells to a state refractory to virus hijacking. In contrast to previous host cell-centric approaches aimed at targeting single host cell proteins that directly interact with the viral proteome, the ViroTreat model was designed to target the entire MR protein module, whose concerted regulatory activity is responsible for implementing and maintaining a virus replication-permissive transcriptional state in the host cell. Thus, ViroTreat expands the one disease/one target/one drug paradigm to targeting an entire protein module (i.e, *Viral Checkpoint*) based on the accurate assessment of each drug's proteome-wide MoA, as dissected from perturbational profile data. Such a holistic approach to matching disease dependencies to drug MoA overcomes the inherent limitations of drug repurposing efforts that focus on

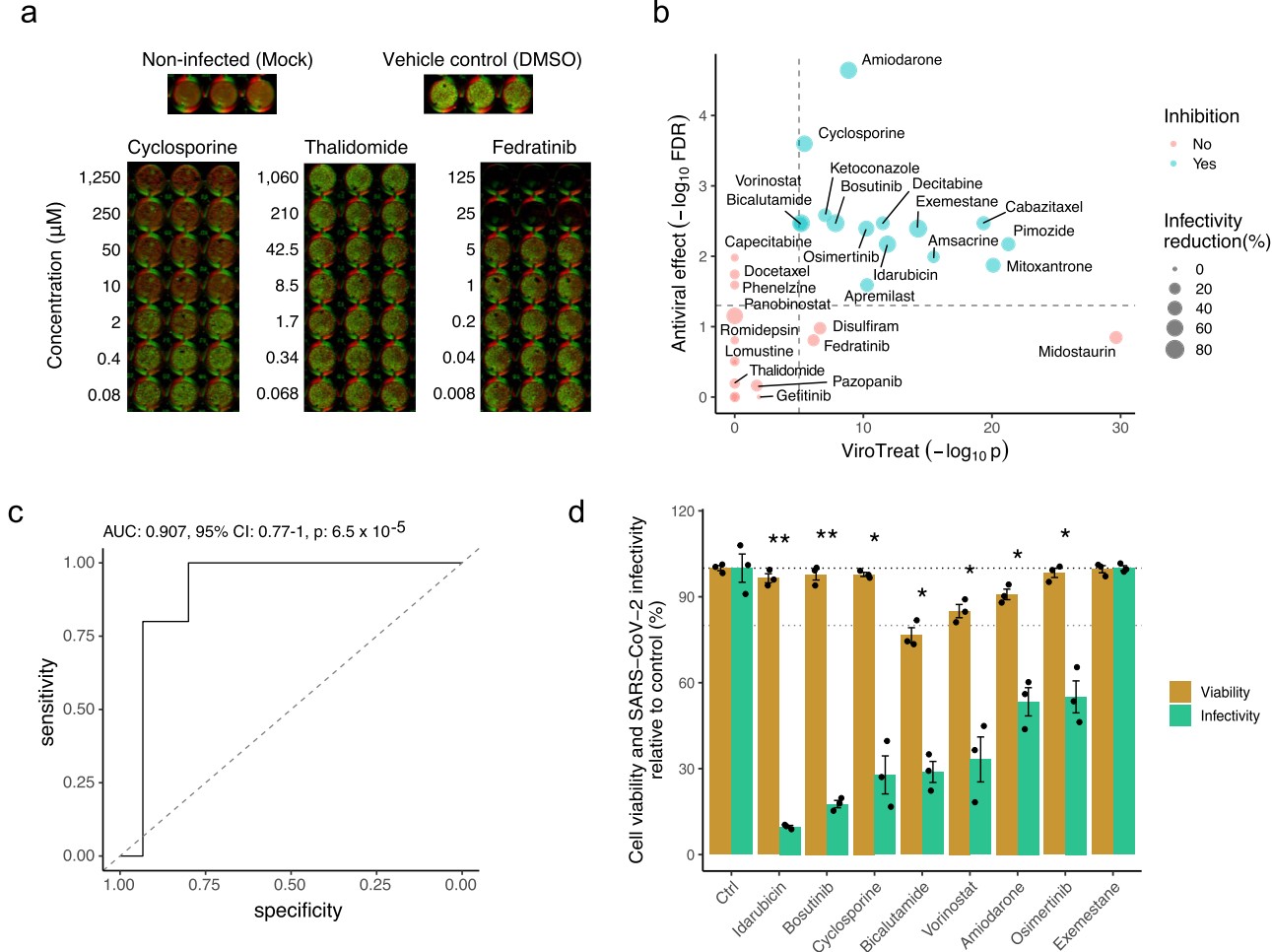

**Fig. 4 Experimental validation of ViroTreat predictions. a** Representative immunofluorescence images of non-infected (Mock) Caco-2 cells, vehicle control (DMSO) treated and SARS-CoV-2 infected cells, and representative examples of a drug showing significant antiviral effect (Cyclosporine), of a drug showing non-significant antiviral effect (Thalidomide) and a drug showing non-significant antiviral effect and cell toxicity (Fedratinib). Drug concentration (μM) is indicated to the left of the images showing triplicated experiments. Cells were stained with DNA dye Draq5 (red) and anti-dsRNA antibody (green). **b** Scatterplot showing the ViroTreat results (x-axis) compared to the specific antiviral effect (y-axis) experimentally evaluated in Caco-2 colon adenocarcinoma cells. The vertical and horizontal dashed lines represent the thresholds for statistical significance for ViroTreat ($p$-value = $10^{-5}$, BC) and specific antiviral effect (FDR = 0.05), respectively. **c** ROC analysis for the ViroTreat predictions, considering as positive response a specific antiviral effect at FDR < 0.05 with at least 20% reduction in virus replication. Estimated AUC, 95% confidence interval (CI) and $p$-value are indicated in the plot. **d** Effect of 8 drugs, showing the strongest reduction in SARS-CoV-2 replication in Caco-2 cells, on cell viability and SARS-CoV-2 replication in GI organoid-derived 2D primary cell cultures. Bars indicate the mean ± SEM. Antiviral effect: *FDR < 0.05, **FDR < 0.01. Source data in Supplementary Data 4 and 5.

inhibitors of individual proteins or single pathways to thwart viral replication as part of a host cell-targeting strategy.

Viral Checkpoint MR identification requires availability of gene expression signatures of virus-infected cells. Therefore, to avoid model-specific confounding effects and to identify a more universal and reproducible MR signature of viral infection, we performed MR analysis in multiple, complementary cellular models, including both transformed cell lines and normal 3D-organoid cultures representing both airway and GI epithelium lineages. In addition, to avoid confounding effects from a heterogeneous combination of infected and non-infected cells—representing the majority of the cell population—MR analysis was also performed at the single cell level, using SARS-CoV-2 genome mapped reads to unequivocally identify infected cells. Moreover, we avoided confounding effects from single cell transcriptional state heterogeneity by comparing each infected cell to a small pool of the closest non-infected cells, based on MR analysis, as controls (Supplementary Fig. 1). Finally, to achieve cell context-

specific elucidation of drug MoA, we analyzed drug perturbations in cell lines that recapitulate the biology of the infected cells, based on conservation of their most differentially active/inactive MRs, as previously described[30].

The ViroTreat framework prioritizes drugs from a predefined library used to generate perturbational assays. For this proof-of-concept, we maximized the translational potential of drug predictions, by focusing our analysis on FDA-approved drugs used primarily in an oncology setting; with particular emphasis on orally available drugs. However, the approach can be easily extended to explore a much larger library of pharmacological compounds. Moreover, the database of drug context-specific MoA can be generated independently and prior to the identification, isolation, and characterization of a viral pathogen of interest, making it readily available for current as well as future pandemics.

In addition, while most studies have focused on drugs that act as high affinity inhibitors of target proteins[2-6,36,37], to our

knowledge, this is the first study to focus on pharmacologic agents predicted to activate, rather than inhibit, an entire protein module of Master Regulator proteins whose inactivation by the virus was found to be necessary for viral hijack and replication. By inducing drug-mediated reversion of the Viral Checkpoint activity, we successfully reprogrammed host cells to a regulatory state of "viral contraception," thereby significantly buffering the virus's ability to hijack the host cell machinery required for its infective cycle.

Critically, Virotreat predicted SARS-CoV-2-specific antiviral activity of drugs that have recently emerged as potential host cell-targeting antivirals, in completely unbiased fashion. Among these, cyclosporine[38], amiodarone[39], pimozide[40], mitoxantrone[41], osimertinib[42], bosutinib[43], and bicalutamide[44]. Moreover, three of the Virotreat-predicted drugs—cyclosporine (NCT04492891), amiodarone (NCT04351763), and bicalutamide (NCT04509999)—are being evaluated in clinical trials for their safety and efficacy in persons with SARS-CoV-2 infection. This further validates the predictive power of Virotreat to define novel antiviral therapeutic approaches and strongly suggest that host cell targeting provides a viable strategy to complement viral-protein targeting drugs.

Among the methodological limitations, the most critical one is the need to obtain physiologic models to identify appropriate infection signatures, generate relevant drug perturbational profiles, and validate predicted drugs. Assessing the optimal concentration at which each compound should be profiled represents an additional challenge. Moreover, lack of available drug perturbational data in suitable lung epithelial models required that we perform the proof-of-concept evaluation in the GI-context, for which drug perturbational data, on a cell line recapitulating the viral checkpoint MRs, is available from the PanACEA collection[32]. While limiting somewhat the immediate translational potential to the clinic, the use of GI-models does not impair the validity of our proof-of concept evaluation and also allowed us to test the specificity of drug-mediated viral checkpoint abrogation for coronavirus vs. rotavirus infection.

In this study, we propose that drug-mediated activation or stabilization of critical MR proteins inactivated by viral infection is the principal mode of action blunting virus replication. Although this mode of action may be inferred from our model, one limitation of the study is lack of direct experimental evidence confirming that reactivation of such specific MRs is the mechanism mediating drug-induced effects on infectivity in the experimental setting. In this regard, it should be noted that when such a model has been applied in the oncology setting, drugs predicted to inhibit tumor growth do so in association with the expected inversion of MRs in the tumor checkpoint activity pattern in vivo[34,45]. However, confirming inversion of critical MRs in our virus-based model presents a number of technical hurdles that require intensive optimization. The most challenging aspect is the difficulty identifying infected cells, given the extremely limited number of copies of the SARS-CoV-2 genome available for analysis when viral replication is significantly inhibited by drug exposure. As a result, designing, optimizing technical features, and performing such experiments based on scRNA-Seq analysis are beyond the current scope of this research effort. However, further investigation of this aspect of the model is certainly warranted.

From a translational perspective, in the setting of both the current and future pandemics, as well as for recurrent epidemics such as those caused by influenza and other viral pathogens, the Viral Checkpoint framework can leverage bulk and single-cell profiles from infected cells to quickly identify the precise set of MR proteins responsible for creating a virus infection-friendly environment in the host cell. Once identified, independent of the specific viral pathogen, potential therapeutic agents can be efficiently prioritized by the ViroTreat model, using readily available—and relatively inexpensive—perturbational databases to elucidate context-specific, proteome-wide drug MoA. Host cell-directed therapies shown to be effective in cell line and organoid models based on such predictions can then undergo rapid validation in more physiologic contexts, prior to testing in human trials designed to evaluate their safety and therapeutic value in the clinical setting.

## Methods

**Cells**. Vero E6 (ATCC CRL-1586) and Caco-2 (ATCC HTB-37) cells were maintained in DMEM supplemented with 10% fetal bovine serum and 1% penicillin/streptomycin.

**GI organoids**. Human tissue was received from colon resection from the University Hospital Heidelberg. This study was carried out in accordance with the recommendations of the University Hospital Heidelberg with informed written consent from all subjects in accordance with the Declaration of Helsinki. All samples were received and maintained in an anonymized manner. The protocol was approved by the "Ethics commission of the University Hospital Heidelberg" under the protocol S-443/2017. Stem cells containing crypts were isolated following previously described protocols[46]. Organoids were passaged and maintained in basal and differentiation culture media (Supplementary Table 2) as previously described[46].

**Viruses**. SARS-CoV-2 (strain BavPat1) was obtained from the European Virology Archive. The virus was amplified in Vero E6 cells and used at a passage 3 for all experiments as previously described[33,47].

**SARS-CoV-2 infection assay**. 20,000 cells were seeded per well into a 96-well dish 24 h prior to drug treatment. 100 μL of media containing the highest drug concentration was added to the first well. Six serial 1:5 dilutions were made (all samples were performed in triplicate). Drugs were incubated on cells for 24 h. Prior to infections, fresh drugs were replaced and SARS-CoV-2 at multiplicity of infection (MOI) 3 (as determined in Vero cells) was added to each well. In these conditions, 70–90% of Caco-2 cells were found infected by SARS-CoV-2, at 24 h post-infection (hpi), in the absence of drugs. 24 h post-infection cells were fixed in 4% paraformaldehyde (PFA) for 10 mins at room temperature (RT). PFA was removed and cells were washed twice in 1X PBS and then permeabilized for 10 mins at RT in 0.5% Triton-X. Cells were blocked in a 1:2 dilution of Li-Cor blocking buffer (Li-Cor) for 30 mins at RT. Cells were stained with 1/1000 dilution anti-dsRNA (J2, SCIONS) for 1 h at RT as marker of infected cells as previously described[33]. Cells were washed three times with 0.1% Tween in PBS. Secondary antibody goat anti-mouse IR 800 (Thermo) and DNA dye Draq5 (Thermo) were diluted 1/10,000 in blocking buffer and incubated for 1 h at RT. Cells were washed three times with 0.1% Tween/PBS. Cells were imaged in 1X PBS on a LICOR imager. Effect of drugs were analyzed by comparing the average fluorescence of mock treated cells to drug treated cells. Draq5 staining was used to determined cell viability.

**Rotavirus infection assay**. 40,000 cells were seeded per well into a collagen-coated 96-well dish 24 h prior to drug treatment. 100 μL of media containing the highest drug concentration was added to the first well. Six serial 1:5 dilutions were made (all samples were performed in triplicate). Drugs were incubated on cells for 24 h. Media was removed and cells were washed 2X with serum-free media and were infected with WT SA11 Rotavirus expressing mKate at MOI 0.1 (calculated in MA104 cells) diluted in serum-free media. Rotavirus was previously activated for 30 min at 37 °C in serum-free media containing 2 μg/ml trypsin. Infection was allowed to proceed for 1 h. In these conditions, 70–90% of Caco-2 cells were found infected by rotavirus, 24 hpi, in the absence of drugs. Following infection, virus was removed and cells were washed 1X with serum-free media. Media containing drugs and 0.5 μg/ml trypsin were added back to cells to allow for Rotavirus propagation. 24 h post-infection cells were fixed with 2% PFA for 15 mins and then stained with DAPI. Cells were imaged in 1X PBS on a Cell Discoverer 7 using a 5X objective. Quantifications of infection was carried out by quantifying the number of infected cells (mKate positive cells) in infected and not infected samples using CellProfiler.

**SARS-CoV-2 infection of human colon organoids-derived 2D primary cell cultures**. Organoids were cultured in 24-well plates in basal medium for 5–7 days following the original protocol of Sato and co-workers[46]. To obtain human colon organoids-derived 2D primary cell cultures, the medium was removed from the 24-well plates, organoids were washed 1X with cold PBS and spun (450 g for 5 mins). PBS was removed and organoids were digested with 0.5% Trypsin-EDTA (Life technologies) for 5 mins at 37 °C. Digestion was stopped by addition of serum containing medium. Digested-organoids were spun again at 450 g for 5 mins and the supernatant was removed and digested organoids were re-suspended in basal

media at a ratio of 250 μL media/well (corresponding to approximately 400 organoids per ml). Prior seeding, the 48-well tissue culture plates were coated with 2.5% human collagen in water for 1 h at 37 °C. The collagen mixture was removed from the 48-well plate and 250 μL of trypsin-digested organoids (corresponding to about 100 digested organoids) were added to each well. 48 h post-seeding differentiation media (Supplementary Table 2) was added to cells and 4 days post-differentiation cells were treated with drugs at the indicated concentrations for 2 h prior to SARS-CoV-2 infection. Media containing drugs was removed and $10^6$ focus forming units (FFU) (as determined in Vero cells) of SARS-CoV-2 was added to each well for 1 h at 37 °C. Following 1 h incubation, virus was removed and fresh differentiation media containing drugs was added to cells. 24 h post-infection RNA was harvested, and virus replication was monitored by RT-qPCR.

**Estimation of the antiviral effect**. We define the antiviral effect of a drug as its viability-normalized effect on SARS-CoV-2 replication. The antiviral effect was quantified as the log-ratio between virus replication and cell viability reduction relative to vehicle-treated controls. Statistical significance was estimated by Student's t-test for each evaluated drug concentration, and multiple-hypothesis testing due to the multiple evaluated concentrations was corrected using the conservative Bonferroni's method. Multiple hypothesis testing due to multiple evaluated drugs was further corrected by Benjamini-Hochberg False Discovery Rate (FDR).

Drugs predicted by ViroTreat were considered validated when showing a significant antiviral effect (FDR < 0.05) and a reduction in virus replication of at least 20%. This additional criterium was used to increase the stringency when evaluating the predictions and the threshold was inferred by fitting a gaussian mixture model (GMM) to the relative replication in response to all evaluated drugs (Supplementary Fig. 9). This analysis identified four groups of drugs—i.e. components of the GMM analysis. The first two groups, based on their mean, showed an average decrease in infectivity of 65% and 30%; the third group showed an average decrease in infectivity close to zero (3.5%); and the forth group showed an average increased in infectivity of 29% (Supplementary Fig. 9). Based on this analysis, we empirically estimated 20% as a reasonable threshold distinguishing drugs that inhibit viral replication (first and second groups) from drugs that showed no effect or increased replication (third and fourth groups, see Supplementary Fig. 9). The GMM analysis was performed using the mixtools package available on CRAN (https://cran.r-project.org/web/packages/mixtools/index.html) (Supplementary Fig. 9).

**RNA isolation, cDNA, and RT-qPCR**. RNA was harvested from cells using RNAeasy RNA extraction kit (Qiagen) as per manufacturer's instructions. cDNA was made using iSCRIPT reverse transcriptase (BioRad) from 250 ng of total RNA as per manufactures instructions. RT-qPCR was performed using iTaq SYBR green (BioRad) as per manufacturer's instructions, TBP or HPRT1 were used as normalizing genes. See Supplementary Table 3 for primers used.

**VIPER analysis of bulk RNA-Seq datasets**. The source for all the datasets is listed in Supplementary Table 1. RNA-Seq raw-counts data for Calu-3, H1299 and Caco-2 cell line models were obtained from Gene Expression Omnibus Database (GEO, GSE148729)[18]. Raw-counts data for A549 cell line, Normal Human Bronchial Epithelial (NHBE) primary cells, a post-mortem lung tissue sample from a COVID-19 patient and a healthy human lung biopsy were downloaded from GEO (GSE147507)[48]. Normalized data (Transcript per Kilobase Million, TPM) for lung organoids were downloaded from GEO (GSE160435). Raw-count data was normalized using the variance stabilization transformation (VST) procedure as implemented in the DESeq package from Bioconductor[49].

Differential gene expression signatures for the Wyler's dataset[18] (GSE148729) were computed by comparing the SARS-CoV-2 infected samples against the centroid—i.e. the average expression of each gene—of the closest matched non-infected (mock) samples as identified by unsupervised clustering. Specifically, we first performed K-means cluster analysis of the normalized gene expression profiles. The optimal number of clusters was estimated by silhouette-score analysis as implemented in the "fviz_nbclust" function of the "factoextra" package (https://cran.r-project.org/web/packages/factoextra/index.html). Cluster solutions were evaluated from $k = 2$ to $k = 10$ and the solution with the highest average of silhouette score was considered as optimal. Based on the optimal cluster solution, we selected as reference for each infected sample the centroid of the mock samples within the same cluster. In cases of clusters constituted by infected samples only, the centroid of the mock samples in the closest cluster were used as reference. Because a two clusters solution was estimated as optimal for all cluster analysis, the other cluster was the trivial closest cluster solution in all cases. Cluster solutions with less than two samples per cluster were considered ineffective. For Calu-3 cell line, we noticed that samples associated to the two series (series-1 and series-2) clustered separately—i.e. samples clustered according to series memberships. To avoid possible batch effects in the analysis, the samples of these two series were re-clustered separately to identify the best matched mock control samples in each series independently. For series-1, the mock samples at 4 h and 24 h clustered together and were used as reference to compute the differential expression signatures of all the Calu-3 SARS-CoV-2 infected samples. For series-2, three mock samples, including one mock sample at 4 h and two mock samples at 12 h clustered

together and were used as reference to compute the differential expression signatures for all the Calu-3 SARS-CoV-2 infected samples. Of note, in series-2, one mock sample at 4 h (GSM4477923) clustered separately from all the other samples with a silhouette score of zero which indicates no clear cluster assignment. This sample was considered as outlier and excluded from the downstream analysis. For the Caco-2 cell line, the centroid of the 4 h mock samples was used as reference to compute the differential expression signatures of the SARS-CoV-2 infected samples at 4 h and 12 h, while the centroid of 24 h mock samples was used as reference to compute the differential expression signatures of the 24 h SARS-CoV-2 infected samples. For the H1299 cell line, the centroid of the 4 h mock samples was used as reference to compute the differential expression signatures of the SARS-CoV-2 infected samples at 4 h and 12 h; and the centroid of the 36 h mock samples was used as reference to compute the differential expression signatures of the 36 h SARS-CoV-2 infected samples.

Differential gene expression signatures for the Blanco-Melo's dataset[48] (GSE147507) were computed using the centroid of the matched—i.e. same cell line or primary cells—mock control samples as reference. For the post-mortem human lung sample from a COVID-19 patient, the differential gene expression signature was computed using the healthy human lung biopsy samples as reference.

Differential gene expression signatures for the lung organoid sample was computed using as reference its matched mock control sample.

The differential activity of 5,734 proteins, including 1,723 transcription factors, 630 co-transcription factors, and 3,381 signaling proteins, was estimated for each of the differential gene expression signatures with the VIPER algorithm[9], using matched context-specific models of transcriptional regulation. Lung, colon and rectal adenocarcinoma context-specific models of transcriptional regulation were reverse-engineered, based on 517 lung, 459 colon and 167 rectal adenocarcinoma samples in The Cancer Genome Atlas (TCGA) with the ARACNe algorithm[10,50], as discussed in ref. [16]. While, ideally, regulatory networks from non-cancer-related epithelial cells may have been more appropriate, use of cancer-related regulatory networks is justified by the high conservation of protein transcriptional targets in cancer-related and normal cells from the same lineage[11]. The regulatory networks are available as part of the aracne.networks R package from Bioconductor. Specifically, protein activity signatures in response to SARS-CoV-2 infection of the lung adenocarcinoma cell lines (Calu-3, H1299 and A549), lung organoids and human lung tissue samples were inferred with the VIPER algorithm using the lung adenocarcinoma context-specific network. Protein activity signatures for Caco-2 colorectal carcinoma cell line were estimate with the metaVIPER algorithm[13] using the colon and rectal adenocarcinoma context-specific networks.

The VIPER-inferred protein activity signatures of infected samples at the same time point in the same cell line were integrated using the Stouffer method[51].

**VIPER analysis of scRNA-Seq datasets**. Single-cell (sc)RNAseq count matrices, based on Unique Molecular Identifiers (UMI), for Calu-3 and H1299 lung adenocarcinoma cell lines were downloaded from GEO (GSE148729). Both count matrices were already filtered for low quality cells as described[18]. Count matrices (UMI) from ileum and colon organoids were made available by Boulant lab and are also publicly available on GEO (GSE156760). Count matrices were filtered for low quality cells as described by Triana et al.[47].

In contrast to bulk RNASeq profiles, single cell RNASeq profiles (scRNASeq) allow effective identification of the individual cells likely to be infected by the virus, which commonly represent a minority of cells in a culture. For this study, therefore, we defined cells to be infected if they present at least one sequenced read mapped to the SARS-CoV-2 genome. Critically, gene expression signatures based on scRNASeq profiles, as computed by comparing bona fide infected cells to non-infected controls, are less affected by contamination and dilution effects characteristic of bulk RNASeq-derived signatures, resulting from a variable proportion of infected vs. non-infected cells.

To account for confounding effects and gene expression profile heterogeneity associated with mechanisms that are independent of viral infection[18,47]—such as cell cycle and the use of models derived from cancer cell lines[52]—differential expression signatures between infected and non-infected single cells were computed by comparing each infected cell to its $k = 50$ closest non-infected ones (Supplementary Fig. 1). This approach significantly improved accuracy and reproducibility of differential gene expression signatures, including across different cell lines, by minimizing confounding effects not associated with viral infection. To identify mock controls cells for each individual infected cell we transformed the count matrices to count per million (CPM) and subsequently to VIPER-inferred protein activity signatures. Briefly, gene expression profiles were transformed to differential gene expression signatures using the "scale" method—i.e. z-score transformation—as implemented in the VIPER package[9]. Then, using lung adenocarcinoma context-specific models of transcriptional regulation, we transformed the single-cell gene expression signature matrices for Calu-3 and H1299 cell lines to VIPER-inferred protein activity signature matrices. Similarly, using colon and rectal adenocarcinoma context-specific networks, we transformed the single-cell gene expression signature matrices for ileum and colon organoids to the corresponding metaVIPER-inferred protein activity signature matrices.

The phenotypic state similarity between cells of the same dataset was quantified by the euclidean distance, calculated based on the top 100 principal components of the VIPER-inferred protein activity matrix. Briefly, the Singular Value

Decomposition (SVD) was used to estimate the matrix of cells by eigenproteins (principal components), and linear regression analysis was used to identify the components (eigenprotein vectors) significantly associated to the viral infection, expressed as the sum of the normalized UMI viral counts—counts mapping to the SARS-CoV-2 genome. For ileum and colon, the vectors of viral counts were generated by summing the normalized counts generated by targeted sequencing analysis[47]. Principal components significantly associated with infection ($p < 0.05$) were removed from the PCA space. Next, we performed a K-Nearest Neighbors (KNN) analysis in the dimensionally reduced PCA space, considering the top 100 infection-independent principal components, to identify the phenotypically closest 50 mock cells for each of the infected cells. The KNN analysis was performed using the FNN package[53]. The 50 phenotypically closest mock cells were used as reference to compute the SARS-CoV-2-induced differential gene expression signature for each of the infected cells. Specifically, the differential gene expression signature for each infected cell was estimated by subtracting the mean expression of the 50 phenotypically closest mock cells and dividing by their standard deviation. For Calu-3 and H1299 cell lines, we considered as "SARS-CoV-2-infected" all the cells with at least 1 sequencing read mapping to the SARS-CoV-2 genome. For ileum and colon, we considered as "SCOV2-infected", all cells identified by targeted sequencing[47].

The differential gene expression signatures of SARS-CoV-2 infected cells were transformed to inferred protein activity signatures by VIPER and metaVIPER algorithms, as described above.

Single-cell protein activity signatures of each data set were integrated by arithmetic mean at each available time point for each cell line.

**Similarity of VIPER-inferred protein activity signatures**. The conservation of MR proteins between VIPER-inferred protein activity signatures was quantified by the reciprocal enrichment of the top 25 most activated, and the top 25 most inactivated proteins in signature $S_1$ in proteins differentially active in signature $S_2$ and vice versa[54], as implemented by the *viperSimilarity()* function in the viper package from Bioconductor.

**Enrichment of biological hallmarks on SARS-CoV-2 infection-induced protein activity signatures**. Hallmarks gene sets (v.7.2) were downloaded from the molecular signatures database (MSigDB) website (http://www.gsea-msigdb.org/gsea/msigdb/collections.jsp). Enrichment of the MsigBD biological hallmarks protein-sets on the SARS-CoV-2 induced, VIPER-inferred protein activity signatures, were estimated with the aREA algorithm[9].

**Enrichment of viral checkpoint MRs on infection essential genes identified by CRISPR screens**. CRISPR screen results (z-score) were downloaded from the supplementary data of Wei et al.[6] (Vero-E6 cells) and Schneider et al.[4] (Huh-7.5 cells). Z-scores were integrated across all experimental conditions for each cell line using the Stouffer's method. Enrichment of the top 50 most activated, and the top 50 most inactivated proteins in response to SARS-CoV-2 infection, obtained after integrating (average) all 10 single-cell protein activity signatures, on each CRISPR experiment z-score signature, and on their Stouffer's integration, were estimated by GSEA. Normalized Enrichment Score (NES) and p-value were estimated by permuting the genes in the CRISPR signatures 10,000 times uniformly at random. SARS-CoV-2 inactivated MRs essential for infectivity were identified as the genes in the leading-edge for the GSEA of the inactivated MRs on the integrated CRISPR screen signature.

**Enrichment of SARS-CoV-2 interacting proteins on host proteins differentially active in response to SARS-CoV-2 infection**. A list of 332 SARS-CoV-2 interacting proteins was obtained from Gordon et al.[2]. 90 of the 332 interacting proteins were represented among the regulatory proteins for which we could infer their activity. Enrichment analysis of this 90 SARS-CoV-2 interacting proteins on the VIPER-inferred protein activity signatures was performed by GSEA. NES and p-values were estimated by permuting the VIPER-inferred protein activity signatures 10,000 times uniformly at random.

**ViroTreat analysis**. Based on the successful outcomes observed with OncoTreat when evaluated in the context of tumor suppression, we sought to develop a novel, analogous algorithm, ViroTreat, to identify small molecule compounds capable of suppressing viral infection by targeting the Viral Checkpoint module. Similar to its use in cancer, ViroTreat systematically assesses and prioritizes a small-molecule compound's ability to reverse the activity of a set of MR proteins based on large-scale drug perturbation assays in cell lines that recapitulate (*a*) the regulatory model of the target cellular population and (*b*) the activity of MR proteins. Specifically, perturbational assay data are comprised of RNASeq profiles generated at 24 h (by PLATE-Seq assays[35]), following treatment of MR-matched cell lines with a library of FDA-approved and late-stage experimental drugs (in Phase 2 and 3 clinical trials) and DMSO as control. These profiles are then used to assess the differential activity of relevant MRs in drug vs. DMSO-treated cells. Finally, enrichment of MR proteins in proteins whose activity has been inverted by the drug is computed by protein set enrichment analysis (PSEA) using the aREA algorithm[8,55]. The RNASeq profiles used for ViroTreat analysis were generated at 24 h following treatment of LoVo cells with a repertoire of 154 FDA-approved

oncology drugs. Perturbations were performed at each drug's highest sublethal concentration (48 h $IC_{20}$) or maximum serum concentration ($C_{max}$) at its Maximum Tolerated Dose (MTD), whichever was lower. This was done to prevent confounding effects, unrelated to the drug MoA, resulting from cell death or stress pathway activation. RNASeq data was generated using PLATE-Seq, a fully automated, 96-well based assay[35] (Supplementary Data 2).

**Statistics and reproducibility**. The number of replicates for each experiment is indicated in the figures. Individual data points and mean ± standard error of the mean (SEM) are shown for bar graphs. Data analysis was performed with R-system v4.0.5. P-values for the drugs' antiviral effect was estimated by Student's *t*-test. Normalized enrichment scores (NES) and P-values for enrichment analysis were estimated by aREA[9] or GSEA[20] with permutation test. Multiple hypothesis tests were addressed by Bonferroni's or Benjamini-Hochberg False Discovery Rate (FDR) methods, as indicated.

**Reporting summary**. Further information on research design is available in the Nature Research Reporting Summary linked to this article.

## Data availability

Availability of SARS-CoV-2 host cell RNA-Seq and scRNA-Seq datasets is indicated in Supplementary Table 1. The drug perturbational dataset (PLATE-seq) for the colorectal adenocarcinoma (LoVo) model is available from ref.[32]. The context-specific interactomes are available from Bioconductor as part of the aracne.networks package for R (https://www.bioconductor.org).The source data for the plots are available as Supplementary Data 4 and 5.

## Code availability

ViroTreat R source code is available in Supplementary Data 3. The aREA algorithm is available from Bioconductor as part of the viper package for R.

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

## Acknowledgements

We would like to thank Dr. Vibor Laketa and Dr. Sylvia Olberg for their support through the Infectious Diseases Imaging Platform (IDIP) and Tatiana Alvarez for original artwork. This research was supported by the following NIH grants to A.C.: R35 CA197745 (Outstanding Investigator Award); U01 CA217858 (Cancer Target Discovery and Development); S10 OD012351 and S10 OD021764 (Shared Instrument Grants); by grants to S.B.: Deutsche Forschungsgemeinschaft (DFG) project numbers 415089553 (Heisenberg program), 240245660 (SFB1129), 278001972 (TRR186), and 272983813 (TRR179), and from the state of Baden Wuerttemberg (AZ: 33.7533.-6-21/5/1) and the Bundesministerium Bildung und Forschung (BMBF) (01KI20198A) and within the Network University Medicine - Organo-Strat COVID-19; by grants to M.L.S.: BMBF (01KI20239B) and DFG project 416072091; by grant to T.A.: ERC Consolidator grant METACELL (grant number 773089); by grant to M.K.J.: BCPM grant to Thomas Geiser (Department of Pulmonary Medicine, University Hospital/DBMR), and support from the Department of Urology of the Bern University Hospital; and to M.J.A.: research support from Karyopharm Therapeutics, Inc.

## Author contributions

P.L., G.B., M.K.J., S.B., A.C., and M.J.A. conceived this work. M.L.S., S.T., P.D., T.A., F.L.M, and M.D.M performed experiments. C.K., R.B.R. and S.P. performed experiments and generated the drugs' perturbational data. P.L., X.S., and M.J.A. performed analysis. G.B., C.K., T.A., M.K.J, A.C., S.B., and M.J.A. supervised experiments and data analysis. P.L., M.L.S., G.B., A.C., S.B., and M.J.A. wrote the manuscript. P.L., M.L.S., G.B., M.K.J., A.C., S.B., and M.J.A. reviewed the manuscript. All authors approved the final manuscript.

## Competing interests

The authors declare the following competing interests: P.L. is Director of Single-Cell Systems Biology at DarwinHealth, Inc., a company that has licensed some of the algorithms used in this manuscript from Columbia University. G.B. is founder, CEO and equity holder of DarwinHealth, Inc. X.S. is Senior Computational Biologist at DarwinHealth, Inc. A.C. is founder, equity holder, and consultant of DarwinHealth Inc. M.J.A. is CSO and equity holder of DarwinHealth, Inc. Columbia University is also an equity holder in DarwinHealth Inc. The remaining authors declare no competing interests.
