## [Peer Review File · Communications Biology]

Reviewers' comments:

Reviewer #1 (Remarks to the Author):

The authors investigate factors of transcriptional regulations during viral infection, in particular in SARS-CoV-2 infection, as novel target of host directed drug treatment. In this work, they benchmark for viral disease research an approach previously implemented in the cancer field. They first applied the VIPER method on scRNAseq data from SARS-CoV-2 infected cells to identify key regulators named by the authors as "Master Regulators" (MR). Then they implemented the ViroTreat algorithm, calqued on the previously established OncoTreat method, to design and analysis a drug screen identifying MR targeting molecules able to impair viral permissive transcriptional response.

Overall this is a very exciting and well done study that should be published! The concept to re-activate MRs to treat virus infections is exciting. However, a number of questions arose that could further strengthen confidence in the here described algorithm and the concept put forward.

For the identification of MR, the authors used the state-of-the-art method for identification transcriptional regulators VIPER and selected two datasets each representative of the lung and the gut lineages, relevant tissues in context of SARS-Cov-2 pathogenicity. scRNAseq data ensures better data quality than bulk RNAseq. Such approach would greatly gain relevance by analysing Covid-19 patient tissues, but the low availability of such samples justify the restriction to cell line data. The identified MR could be benchmarked on bulk RNAseq data from corresponding cell line lineages, a valid approach as no relevant datasets was yet available when the authors performed the study. However, a study was recently published by Melms et al. on lung scRNAseq from patient with lethal Covid-19 infection, and the VIPER results would greatly profit from being validated from this dataset (<https://www.nature.com/articles/s41586-021-03569-1>).

The authors defined an arbitrary number of differentially activated MR in SARS-CoV-2 infected samples (25 activated and 25 inactivated), and then aimed to characterize their functionality. A proper review of the pertinence of this cut-off would benefit from providing individual MR activity change value with associated statistical test as estimated by the VIPER analysis. Pathway enrichment analysis revealed a functional dichotomy between activated and inactivated MR, with an interesting enrichment for innate immunity transcriptional regulators being inactivated, suggesting that inhibiting the host defense transcriptional response could be a viral evasion mechanism.

Interestingly, the authors identified an enrichment for inactivated MR in targets of SARS-CoV-2 viral proteins as published by Gordon et al. from viral protein AP-MS experiments performed in Human Embryonic Kidney (HEK) 293T cells (Sup. Fig. 3). This suggested the binding of viral proteins to host transcriptional regulator as a molecular mechanism of immune evasion during infection, which is an interesting observation. Since a number of interactome studies using both AP-MS and BioID were published and the data is partially diverging it may be interesting whether similar relationships can be observed in these datasets. Such an analysis would give additional confidence to the conducted analysis. Moreover, since these interactome datasets have been published in different cellular systems it would sort of validate these data, which would be highly informative for the scientific community. Moreover, since elegant data has been published on the inhibitory activity of viral ORFs (e.g. Hayn M, et al., <https://www.sciencedirect.com/science/article/pii/S2211124721004654?via=ihub>), it may be interesting to integrate information from this study and/or similar manuscripts. The main question I have on the provided statement is whether this a general pattern across all viral proteins, which would be rather unexpected, or is it the function of selected, overrepresented viral proteins? This would allow identification of MR-viral protein interactions reproducible across independent studies, and thus which are more likely to be functionally important.

The authors found an enrichment for genes essential for SARS-CoV-2 replication in inactivated MR. It is unclear to me why the authors reversed the enrichment analysis here. Would it not be more coherent to verify if the same population of inactivated MR are enriched in the SARS-CoV-2 essential genes, as performed for the previous host targets enrichment analysis? The pertinence of the datasets chosen here can be discussed: a monkey cell line model (Vero), and a kidney cancer cell line with known defects in innate immune pathways (Huh-7.5). Daniloski et al. (<https://doi.org/10.1016/j.cell.2020.10.030>) performed a genome-wide CRISPR screen in A549-ACE2

cells which should be closer to, at least, the lung model analyzed in this study. Additionally, it would be interesting for the authors to compare their data with the CRISPR screens dataset of Rebendenne et al., yet under review but performed in Caco2 and Calu3 cell lines

<https://www.biorxiv.org/content/10.1101/2021.05.19.444823v2>

To identify drugs which could rescue the MR inhibition and thus limit SARS-CoV-2 replication, the authors applied the ViroTreat algorithm, adapted from the Oncotreat originally established for cancer treatment prediction. In brief, the authors selected the LoVo cells as being the best representative of the gastro-intestinal lineage and the Caco2 cells, from the PanACEA, a database of cancer cell lines drug perturbation RNAseq profiles. No model cells line could be found to analyses the lung lineages model identified. They utilize RNAseq screens of 154 FDA approved drugs in LoVo cells, and used VIPER to identify the MR differentially activated by each treatment. Among these, 22 drugs were selected to activate MR identified previously as inactivated during SARS-CoV2 infection (Supplementary table – ViroTreat score). 18 of these drugs, and 12 control non effective drugs, were further screened in Caco2 cells for their effect on SARS-CoV-2 replication, among whose 15 of the candidates and none of the control showed an effect (at least 20% reduction) on SARS-CoV-2 replication. The definition of the 20% threshold is clearly enunciated in the method part. Notably, the authors verified the SARS-CoV-2 specificity of the identified drugs by testing their effect on rotavirus infection, another RNA virus with a strong tropism for the gastro-intestinal lineage. This is an interesting and important control. However, the main component contributing to MRs identified in SARS-CoV-2 infections related to innate immune pathways. Since rotavirus has been reported to be sensitive to interferon and to induce innate immune pathways it would be important to use the here established pipeline to show that rotavirus is inducing a different subset of MRs and therefore identified drugs. Since transcriptome data is available for rotaviruses, this should be possible. This would improve the confidence in the here described drug identification pipeline and validate its applicability.

The authors propose that the re-activation of MRs can be used to curb SARS-CoV-2 replication. This is a very interesting concept. However, I feel that this statement needs to be experimentally verified, which should easily be possible.

Overall the authors present an innovative approach to identify novel host directed antiviral drugs, with a focus on Sars-CoV-2 infection, from which the community would certainly profit. They address important questions regarding analysis benchmarking, and validation in various cellular models, as well as non-cancer models. I support publication of this manuscript with some points addressed:

Specific points

- The MR identification by VIPER would profit from a benchmarking on a recent dataset from Melms et al., providing lung scRNAseq from patient with lethal Covid-19 infection: <https://www.nature.com/articles/s41586-021-03569-1>.
- The authors provide ranking analysis for main affected MRs. The manuscript would benefit if the authors provided the individual MR activity difference value with an associated statistical test as estimated by the VIPER analysis (if this is possible). They should report these data at least for the MR highlighted by the different enrichment analysis (pathways, viral protein binders, SARS-CoV-2 replication essential genes).
- The authors use information from CRISPR/Cas9 screens performed in Vero and Huh-7.5 cells to identify essential genes for SARS-CoV-2 replication. Given the availability of other datasets that are closer to the model tested (e.g. Daniloski et al., <https://doi.org/10.1016/j.cell.2020.10.030> – employing A549-ACE2 cells) it may be beneficial to rather consider this dataset. Other CRISPR screens, e.g. Rebendenne et al., are still under review but were performed in Caco2 and Calu3 cell lines <https://www.biorxiv.org/content/10.1101/2021.05.19.444823v2>.
- Please provide data regarding the comparability of the SARS-CoV-2 and rotavirus infection: since the chosen MOI was very different the authors should provide information on similar infection rates – i.e. are similar numbers of cells infected? Are the viruses replicating similarly over time when choosing this MOI?
- The authors benchmarked the used drugs and mention a success rate of 80% but this value is based on a relatively limited number of tested drugs, particularly the ones that are top-rated in the analysis.

Some of the drug activities are just above the arbitrary used cut-off. Therefore I ask to refrain from giving % success rate estimates and suggest to mention the number of successfully tested versus total tested drugs.

- Based on the obtained signatures the authors strongly propose to activate MRs as mode of action to blunt virus replication. Although this mode of action may be inferred from this analysis, it is not clear to what extent such re-activation is really contributing to the inhibitory effect. Experimental evidence that reactivation of MRs in the experimental settings used needs to be provided. I.e. The authors should provide information on treated infected cells and show reversion of target gene expression for a selected number of candidate drugs tested (e.g. by qPCR of target genes).

Minors points

- Line 91: mtaVIPER \diamond metaVIPER
- Figure 1 caption: "top 10", or top 50?
- Supplementary fig. 1: precise exact gene/protein numbers
- Supplementary fig. 2: duplicate information. The authors should remove one side of the correlation heatmaps for clarity.
- To clarify: Line 189: "2-tailed GSEA" when the Sup Fig. 3 caption precise one-tailed test.
- Since this manuscript is about benchmarking a drug-search algorithm, it would be interesting (but not necessarily required) to show a few numbers on how it compares to other algorithms that were proposed by other laboratories and that are sometimes based on other data, i.e. that are or are not considering RNAseq and drug perturbation information but rather focus on integration of drug targets and network analyses (for instance: Siminea et al., <https://academic.oup.com/bib/article/23/1/bbab490/6447433>; Zhou et al., <https://www.nature.com/articles/s41421-020-0153-3>; Gysi et al., <https://www.ncbi.nlm.nih.gov/pmc/articles/PMC7280907/>, Sadegh et al., <https://www.nature.com/articles/s41467-020-17189-2>). Alternative approaches are already nicely discussed, particularly the clear ranking of drugs, which others often do not offer. However, similar drugs (often many more) were identified in other studies; unfortunately validation of these drugs often appear to have limited success rate (at least in our personal experience). This benefit of the here described algorithm may be further discussed.
- Supplementary fig. 2: inversion of the labels in b-c between the caption and the figure. The text description refers to the caption: "inactivated MRs were significantly more conserved than activated MRs, both across models and lineages". But as it is one of the major point of the authors to define subsequent analysis, the authors should provide the raw data of the similarity of VIPER-inferred activity analysis to verify that there is no mismatch in the interpretation.
- The authors suggest the binding of viral proteins to inactivated MR indicate a host immunity evasion mechanism by the virus. The authors should provide as supplementary data an overview of the identified viral proteins and their targets: is this a general pattern across viral proteins, which would be rather unexpected, or is the reported overlap due to overrepresentation of a few viral proteins? Moreover, the manuscript would benefit if the authors considered to reproduce the inactivated MR enrichment analysis on other interactome datasets.

Reviewer #2 (Remarks to the Author):

Brief summary of the manuscript:

Laise and co-workers have performed a proof-of-concept study to systematically search for and re-purpose existing approved drugs for treatment of SARS-CoV-2 infection using several algorithms originally developed in the field of oncology. Their approach is to activate host proteins (master regulators) that have been inactivated by SARS-CoV-2 in order to make the target cell amenable to viral replication; by reversing this inactivation, the host cell will be less supportive of viral replication. Firstly, they use VIPER to analyse publicly available single cell RNA sequencing data from infected

SARS-CoV-2 epithelial lung adenocarcinoma and gastrointestinal organoid models, to identify candidate master regulator proteins enriched from differentially expressed genes between infective states. They use GSEA to assess enriched hallmark pathways in inactivated and activated master regulator proteins. To justify focusing on inactivated master regulator proteins, the authors use publicly available data from mass spec and CRISPR screens to identify proteins that interact directly with SARS-CoV-2 proteins, and proteins that are essential to viral infection cycle respectively, both of which are found to be enriched for inactivate master regulator proteins. The authors combine usage of a publicly available database of drug perturbation RNA sequencing profiles and their own experimental data to screen and select drugs that are predicted to reverse SARS-CoV-2 inactivation of master regulator proteins. Finally, they assess the effect of drug treatment on cell viability and viral replication.

Overall impression of the work:

The methodology of this proof-of-concept study is justified as several drug candidates identified by the authors are currently in clinical trials for COVID-19 treatment. This approach can be potentially applied to search for novel drugs to treat other viral infections, as the drug targets are host proteins rather than viral proteins. This study provides an important and swift method for searching for novel treatments for SARS-CoV-2, as these drugs are already approved for use in humans; however, there are several issues as listed below.

Specific comments, with recommendations for addressing each comment

1. The authors themselves are aware of the limitations of this approach, where the appropriate physiological cellular models are required to obtain gene signatures between infected and non-infected cells, and for drug selection from drug perturbation databases. The authors initially assess lung and GI-derived cells; however, based on the small selection of cell types from the PanACEA database, proceed to focus only on GI-lineage cells. To this end, the final treatments may be more focused in treating GI infected cells (as highlighted by the authors' subsequent testing on lung adenocarcinoma cells), rather than lung-infected cells, which is the primary site of infection. This results in the exclusion of a large number of lung-specific drugs, and is possibly a limitation of re-purposing an oncology database, and limits the overall usefulness of this study. Would the authors be able to find alternative drug perturbation profile databases that would be more appropriate for this study?

2. The authors test their selected drugs on GI cells by pre-treating cells with the appropriate drug for 24 hours before SARS-CoV-2 infection. Although this prophylactic approach may be clinically useful in the future, current treatments for SARS-CoV-2 are therapeutic (ie given after initial infection), so it would be useful to know if similar results are seen if cells are treated with the desired drug after SARS-CoV-2 infection. It would also be interesting to know why the authors chose LoVo cell line to perform the initial drug screen, and Caco-2 to perform the infectivity assay.

3. The authors seed 20,000 cells/well at 0 hours and then treat with the appropriate drug for 24 hours before replacing with fresh media and infecting with SARS-CoV-2. During this period, would some cell death occur, especially at higher concentrations; therefore would there be the same number of cells across all concentrations at the time of infection? Hence the MOI would be inaccurate if there were different cell numbers. It would be perhaps better to pre-treat the cells in a flask for 24 hours, then re-seed for infection.

4. The authors use immunofluorescence to measure viral infection (anti-dsRNA antibody) and cell viability (Draq5). However, in the methods, the authors fix and permeabilize the cells first (for anti-dsRNA staining), and then stain with Draq5; this would have the effect of staining all host cells, including both live and dead cells. The authors see a drop in "cell viability" at high concentrations which could justify dead cells at high concentrations not binding the Draq5 dye – however, truly dead cells may have rounded up and floated off into the media during initial 24 hour drug treatment, and after fixation and washing. Therefore, Draq5 may capture live and dead cells still stuck to the plate at lower concentrations, but properly dead cells would have been removed and therefore no fluorescence

can be seen at higher concentrations. As the authors clearly state that it is important to find a drug treatment that stops viral replication but keeps cells alive, it would be useful if they could use a stain that specifically measures live cells only, for example Calcein AM which relies on cellular enzymatic activity in live cells for fluorescence, rather than the use of DNA intercalating dyes, which can stain both live and dead cells indiscriminately. This is of particular importance as the authors calculate the antiviral effect based on the log-ratio between viral replication and cell viability reduction.

Minor:

1. The Introduction needs to be revised for clarity and content. There needs to be more detail on what master regulator proteins are for those not familiar with the terminology, and their importance (including relevant references). Certain paragraphs of the Introduction would be more appropriate for the Results and Discussion ie where the authors describe and justify in detail their approach to this proof-of-concept study.

2. Certain sections of the Results would benefit from more detail; for example, in the section titled "SARS-CoV-2 induced MR signature", it is not mentioned that VIPER is the algorithm used to analyse single cell RNA sequencing data. Line 141 states: "Single cell analysis revealed highly conserved differential protein activity signatures, as defined by the top 50 most differentially active candidate MRs ..." – is this analysis conducted using VIPER? Also, it is stated two sections later that "5,734 (proteins) we analysed by VIPER" (line 187) – this is not mentioned in the first section. This could be clarified by moving line 562 in Methods ("The differential activity of 5,734 proteins, including 1,723 transcription factors, 630 co-transcription factors, and 3,381 signaling proteins, was estimated for each of the differential gene expression signatures with the VIPER algorithm ...") to the appropriate Results paragraph. It would also be useful to conclude each Results section with an appropriate summarizing sentence, to help the reader understand and interpret the results stated.

3. Line 142: is there a reason why there is a focus on only the top 50 candidate proteins? How was the number 50 chosen?

4. Please clarify if the findings in line 160 – 162 are from bulk or single cell sequencing as it is not stated here in the Results.

5. Line 218: please add more detail to explain the use and purpose of OncoMatch; this seemed vital in excluding lung lineage lines from further analysis, and so should be expanded upon.

6. Line 237: please provide a short explanation of the aREA algorithm

7. Figure 1a: legend states top 10 genes, but there appears to be more genes shown on heatmap?

8. Figure 1a: figure shows NES color bar but no explanation in legend; does darker colour indicate enrichment or expression/activity?

9. Figure 3: difficult to interpret, would benefit from adding an additional summary sentence in the relevant Results section to aid in understanding this figure, for example after the following sentence in lines 239 – 241: "Among the 154 FDA-approved drugs profiled in LoVo cells, ViroTreat prioritized 22 (13 orally available and 9 intravenous) at a highly conservative statistical threshold ($p < 10^{-5}$, Bonferroni corrected (BC)), see Fig. 3 and Supplementary Table 2)." – therefore can it be concluded that these top 22 drugs are predicted to significantly activate more of the top 50 genes/proteins that are inactivated after SARS-CoV-2 infection?

10. Figure 4a: well reflections make figure hard to interpret.

11. Figure 4b: is it possible that some drugs can show high infectivity reduction but no inhibition? Needs more detail on what is being inhibited (viral replication?)

12. Figure 4c: What does the dotted line show?

13. As mentioned in point 2, certain sections of the Methods could be moved to the Results to aid the reader's understanding of the paper. For example, lines 664 – 667 describing ViroTreat could be moved to the relevant Results paragraph.

14. Methods: RNA isolation, cDNA and RT-qPCR: need to correct spelling of manufacturer's

15. Supplementary Figure 1: figure legend would be more informative if provided with numbered sub-sections 1-6 to match figure workflow.

16. Supplementary Figure 6: is it necessary to show multiple graphs for the same drug eg Amiodarone – if they are showing different concentrations, is it possible to merge the data onto a single graph?

17. Supplementary Figure 9: what do the other colored lines show?

Reviewer #1

Specific points

1. The MR identification by VIPER would profit from a benchmarking on a recent dataset from Melms et al., providing lung scRNAseq from patient with lethal Covid-19 infection: <https://www.nature.com/articles/s41586-021-03569-1>.

We thank the reviewer for the comments and for suggesting the comparison with Melms et al., 2021 data.

Per the review's recommendation, we downloaded the gene expression count data and the meta data related to Melms et al., 2021 publication from GEO database (GEO ID=GSE171524) which, as noted, were generated by single-*nucleus* RNA-seq technology (snRNA-seq); moreover, the UMI reads of each individual nucleus were aligned on both, human and SARS-COV-2 genomes. The data set comprises lung tissue from 19 patients who died from COVID19 and 7 control samples.

Among the 19 COVID-19 patient samples analyzed as recommended, we have been able to identify, based on alignment of sequencing reads to the SARS-CoV-2 genome (# of reads > 0), only very few (25/30,069, corresponding to 0.08%) epithelial SARS-CoV-2 infected cells, and such alignments were identifiable in only in a small subset (5/19, 26%) of patients. We suspect that this is most likely because the snRNA-seq technology aims at sequencing transcripts within the *nucleus*, whereas SARS-CoV-2 genome replication occurs in the *cytoplasm*. This limitation impairs our ability to properly distinguish infected from non-infected cells, a critical requirement for elucidating the SARS-CoV-2- infected host Master Regulator proteins from clinical samples. Unfortunately, because of this limitation, the data extracted from the Melms et. al. study are not suitable for MR benchmarking within the experimental context and techniques of the pipeline/model we report.

Nevertheless, as per request of this reviewer, and because we agree such benchmarking could potentially be valuable, we applied analyses consistent with our pipeline (as described in Supplementary Figure 1) to those 5 patients showing nuclei with sequencing reads mapping to the SARS-CoV-2 genome. This sub-analysis showed that the consensus SARS-CoV-2 inactivated MR proteins, as identified from the cell line models we employed, were indeed significantly conserved in 2 of the 5 analyzed patients ($p < 10^{-5}$, 1-tailed aREA test). However, while these results suggest an interesting convergence or signal, because of the aforementioned limitations in identifying/differentiating infected from non-infected cells based on viral reads in the nucleus, per se, the results obtained using this analysis could be significantly biased and, as such, we feel it's best not to include them in the manuscript.

2. The authors provide ranking analysis for main affected MRs. The manuscript would benefit if the authors provided the individual MR activity difference value with an associated statistical test as estimated by the VIPER analysis (if this is possible). They should report these data at least for the MR highlighted by the different enrichment analysis (pathways, viral protein binders, SARS-CoV-2 replication essential genes).

This is an excellent suggestion. As per the reviewer's request, we now have provided the differential protein activity quantified by VIPER, expressed as Normalized Enrichment Score (NES), for all the models used to infer the SARS-CoV-2 infection Master Regulator Proteins. These results are now provided in Supplementary Table 2. Accordingly, the previous version's Supplementary tables 2, 3 and 4 have been updated to S3, S4 and S5, respectively. We also reference these new Supplementary Table 2 in line 149.

3. The authors use information from CRISPR/Cas9 screens performed in Vero and Huh-7.5 cells to identify essential genes for SARS-CoV-2 replication. Given the availability of other datasets that are closer to the model tested (e.g. Daniloski et al., <https://doi.org/10.1016/j.cell.2020.10.030> –employing A549-ACE2 cells) it may be beneficial to rather consider this dataset. Other CRISPR screens, e.g. Rebendenne et al., are still under review but were performed in Caco2 and Calu3 cell lines <https://www.biorxiv.org/content/10.1101/2021.05.19.444823v2>.

We thank the reviewer for this comment. The CRISPR data generated by Daniloski et al., 2020 were not included in this manuscript because the authors did not provide the z-score values as a measure of essentiality of each gene. Indeed, z-scores values are necessary for integrating CRISPR signatures across data sets by the Stouffer's method. As a consequence, these data would not ideally be integrated with the other CRISPR data included in our manuscript. However, as per request of this reviewer, we performed an independent comparison between SARS-CoV-2 MRs and the CRISPR data provided by the Daniloski et al., 2020 work, as recommended. Because of the lack of z-score values, we computed a CRISPR signature by integrating the Fold Change values computed at the two different MOIs reported in the manuscript. The comparison of this signature, indeed, showed a significant enrichment ($p \sim 0.003$, two-tailed GSEA with permutation test, estimated with 10,000 permutations uniformly at random) with the top 50 most inactivated MR proteins (Figure 1), identified in our revised manuscript as Supplementary Fig. 4g.

Figure1. GSEA plot showing the enrichment of the top 50 inactivated MRs by SARS-CoV-2 on the CRISPR signature ranked based on fold changes (FC) values (x-axis) from left (negative FC) to right (positive FC). Normalized enrichment score (NES) and p-value were estimated by two-tailed GSEA using 10,000 permutations.

Regarding Rebenedenne et al. CRISPR screen, the authors only provide the top hits from their screen results, making it impossible to combine their results with the other screens (based on z-scores) or to estimate the enrichment of the most inactivated MR candidate proteins on the CRISPR screen signature. However, as per this reviewer's request, we estimated the enrichment of the CRISPR hits (antiviral genes) reported by Rebenedenne et al. for Caco-2 and Calu-3 on the complete, consensus vector of SARS-CoV-2-induced differential protein activity, obtained by integrating the differential protein activity signatures along models and time points. Again, using this methodology, we observed a significant enrichment of the antiviral genes, identified by Rebenedenne et al., among the inactivated candidate MR proteins ($p < 0.002$, two-tailed GSEA with 10,000 uniformly random permutations; see Figure 2 and Supplementary Fig. 4h and i in the revised version of the manuscript).

Figure 2. GSEA plots showing the enrichment of the antiviral genes reported by Rebnedenne et al., 2021 in Calu3 (left) and Caco2-ACE2 (right) on the SARS-CoV-2-induced VIPER signature (x-axis) identified in our work. The VIPER signature represents the candidate MR proteins identified by VIPER ranked from the most inactivated (left/blue) to the most activated (right/red). Normalized enrichment score (NES) and p-value were estimated by two-tailed GSEA with 10,000 uniformly random permutations.

In light of these significant corroborations/enrichments for inactivated MRs associated with SARS-CoV-2 infection identified among the CRISPR screen data from these two studies recommended by the reviewer, we have included and noted these findings in the main manuscript and provide the results in Supplementary Fig. 4h-I and in the results section, line 213.

4. Please provide data regarding the comparability of the SARS-CoV-2 and rotavirus infection: since the chosen MOI was very different the authors should provide information on similar infection rates – i.e. are similar numbers of cells infected? Are the viruses replicating similarly over time when choosing this MOI?

We thank the reviewer for this comment. SARS-CoV-2 virus stocks are traditionally titred in Vero cells. Rotavirus stocks are commonly titred in MA-104 cells. In this work, we infect Caco-2 cells with different amounts of SARS-CoV-2 or rotavirus (as titred in Vero and MA-104 cells, respectively) to obtain similar infection levels in Caco-2 cells. In our setup, we aim to obtain between 50-80% infected Caco-2 cells by either SARS-CoV-2 or Rotavirus. Representative pictures of Caco-2 cells infected (in the absence of drugs) with SARS-CoV-2 or Rotavirus are presented below in Figure 3. To clarify this in the manuscript, we have added the following sentence in line 497 and 516. “In these conditions, 70-90% of Caco-2 cells were found infected by SARS-CoV-2/rotavirus, 24 hpi, in the absence of drugs”.

Figure 3. Caco-2 cells were infected with either SARS-CoV-2 or rotavirus. 24 hours post-infection cells were fixed and stained for virus infection. (top panel) SARS-CoV-2 infected Caco-2 cells, red= anti-dsRNA immunofluorescence, blue=DAPI. (bottom panel) Rotavirus infected Caco-2 Cells, green=rotavirus-driven mKate expression, blue=DAPI. Three representative images are shown.

SARS-CoV-2 has a shorter life cycle compared to rotavirus (7-8 hrs compared to around 12 hrs). Importantly, for the analysis and the interpretation of our data, in both viral conditions, cells are pre-treated for 24 hrs with the drugs and infection is allowed for 24 hrs in the presence of drugs. These are performed in multiwell plates with each plate containing non-infected control, non-treated but infected controls. This allows for direct comparison of the impact of the drugs on virus infection of Caco-2 cells.

5. The authors benchmarked the used drugs and mention a success rate of 80% but this value is based on a relatively limited number of tested drugs, particularly the ones that are top-rated in the analysis. Some of the drug activities are just above the arbitrary used cut-off. Therefore I ask to refrain from giving % success rate estimates and suggest to mention the number of successfully tested versus total tested drugs.

In agreement with the reviewer suggestion, we have updated the manuscript in lines 58 and 126.

6. Based on the obtained signatures the authors strongly propose to activate MRs as mode of action to blunt virus replication. Although this mode of action may be inferred from this analysis, it is not clear to what extent such re-activation is really contributing to the inhibitory effect. Experimental evidence that reactivation of MRs in the experimental settings used needs to be provided. I.e. The authors should provide information on treated infected cells and show reversion of target gene expression for a selected number of candidate drugs tested (e.g. by qPCR of target genes).

We agree with the reviewer regarding the relevance of providing confirmatory experimental evidence for the lack of inactivation of the MRs following infection by SARS-CoV-2 in the presence of the drugs prioritized by ViroTreat. We have performed similar experiments to corroborate that the anti-tumor effect of OncoTreat-identified drugs *in vivo* is associated with the inversion of the tumor checkpoint activity pattern. We have seen that most (>90%) of the OncoTreat-predicted drugs showing tumor growth inhibition do so in association with the predicted inversion of the tumor checkpoint activity pattern *in vivo* (please, see <https://www.biorxiv.org/content/10.1101/2021.10.03.462951v2> and <https://www.biorxiv.org/content/10.1101/2022.02.11.479456v1>).

Unfortunately, despite the reviewer's well-reasoned rationale, as well as our own interest in confirming inversion of the down-activated MRs, there are several aspects related to the experimental design that make it extremely difficult, if not impossible, to perform such experiments in the absence of a major long-term effort that is beyond the scope of this research program: (1) Since SARS-CoV-2 infects only a fraction of the cells in the well during the infection experiment, performing bulk expression profile may lead to strongly biased results, in particular through a dilution effect and bias contributed by the transcriptome of by-stander, non-infected cells; (2) Performing the analysis by sc(single cell)RNA-Seq would, in principle, allow for the identification of the infected cells based on the presence of sequencing reads mapping to the SARS-CoV-2 genome. Indeed, we have deployed this approach to elucidate the candidate SARS-CoV-2-induced host MR proteins. However, since the evaluated drugs are effective at inhibiting virus replication (or infectivity)—in agreement with their effect as activators of the host MR proteins being inactivated during virus infection and postulated to be required for the virus to hijack cellular components necessary for its replication—the number of copies of the SARS-CoV-2 genome on those “infected” cells would likely be too low to be detected (we have been able to detect a median of only 16 and 1 viral reads per infected cell in the Calu-3 and H1299 scRNA-Seq datasets, respectively); (3) We currently have biohazard related restrictions to obtain the scRNA-Seq expression profiles of infected cells in the facilities we currently have access to, and performing the analysis by fixed single nucleus (sn)-RNA-Seq is not an option we can consider, since identification of nuclei of infected cells cannot be performed by detecting sequencing reads mapping to the SARS-CoV-2 genome, whose replication takes place in the cytosol.

However, because we believe the reviewer's point is well taken, and that under the most ideal circumstances, it would be relevant to confirm the inferred mechanism-of-

action (MOA)—i.e. reactivation of or preventing inactivation of the critical MR proteins linked to viral infection—we have added the following sentence to the manuscript in line 398 in the limitations sections:

“In this study, we propose that drug-mediated activation or stabilization of critical MR proteins inactivated by viral infection is the principal mode of action blunting virus replication. Although this mode of action may be inferred from our model, one limitation of the study is lack of direct experimental evidence confirming that reactivation of such specific MRs is the mechanism mediating drug-induced effects on infectivity in the experimental setting. In this regard, it should be noted that when such a model has been applied in the oncology setting, drugs predicted to inhibit tumor growth do so in association with the expected inversion of MRs in the tumor checkpoint activity pattern *in vivo*^{34,45}. However, confirming inversion of critical MRs in our virus-based model presents a number of technical hurdles that require intensive optimization. The most challenging aspect is the difficulty identifying infected cells, given the extremely limited number of copies of the SARS-CoV-2 genome available for analysis when viral replication is significantly inhibited by drug exposure. As a result, designing, optimizing technical features, and performing such experiments based on scRNA-Seq analysis are beyond the current scope of this research effort. However, further investigation of this aspect of the model is certainly warranted.”

Minor points

1. Line 91: mtaVIPER \$\diamond\$ metaVIPER

The thank the reviewer for finding this typo.

2. Figure 1 caption: “top 10”, or top 50?

Figure 1a shows the differential activity for the top 10 most activated proteins in each of the 10 experimental conditions evaluated. We have clarified this in the legend.

3. Supplementary fig. 1: precise exact gene/protein numbers

We appreciate the reviewer has pointed this out. We have replaced the approximate numbers by the exact ones in Supplementary Figure 1.

4. Supplementary fig. 2: duplicate information. The authors should remove one side of the correlation heatmaps for clarity.

We have removed the lower triangle of the symmetric correlation heatmaps in Supplementary Fig. 2.

5. To clarify: Line 189: “2-tailed GSEA” when the Sup Fig. 3 caption precise one-tailed test.

We appreciate the reviewer finding this error. The legend for Supplementary Fig. 3 correctly express now that the test used is 2-tailed.

6. Since this manuscript is about benchmarking a drug-search algorithm, it would be interesting (but not necessarily required) to show a few numbers on how it compares to other algorithms that were proposed by other laboratories and that are sometimes based on other data, i.e. that are or are not considering RNAseq and drug perturbation information but rather focus on integration of drug targets and network analyses (for instance: Siminea et al., <https://academic.oup.com/bib/article/23/1/bbab490/6447433>; Zhou et al., <https://www.nature.com/articles/s41421-020-0153-3>; Gysi et al., <https://www.ncbi.nlm.nih.gov/pmc/articles/PMC7280907/>, Sadegh et al., <https://www.nature.com/articles/s41467-020-17189-2>). Alternative approaches are already nicely discussed, particularly the clear ranking of drugs, which others often do not offer. However, similar drugs (often many more) were identified in other studies; unfortunately validation of these drugs often appear to have limited success rate (at least in our personal experience). This benefit of the here described algorithm may be further discussed.

We appreciate this comment and share the reviewer interest regarding the comparison of ViroTreat drug predictions to alternative approaches. However, such comparisons are not straightforward given several limitations related to the available data; in particular, based on the limited number of predicted drugs that have been experimentally tested as inhibitors of virus infectivity or replication, and the lack of a systematic experimental validation approaches. We think that results of such comparisons based on data lacking a systematic evaluation of drugs might lead to misleading results. Moreover, our manuscript is less about benchmarking a drug-search algorithm, than postulating the existence, from a biological/regulatory perspective, of a transcriptional regulatory module (viral checkpoint) whose concerted activity is modulated—i.e., inactivation of MR proteins—during virus infection and is required for and permissive of effective virus replication. We used drugs predicted to activate the MR proteins that are part of the viral checkpoint as an experimental tool to demonstrate its requirement for effective viral replication. The use of ViroTreat as a drug-searching algorithm for translational and clinical research would still require perfecting it, both by using more physiologic models and comprehensive drug libraries, as we discuss in the manuscript related to its current limitations (see revised manuscript lines 388-411).

7. Supplementary fig. 2: inversion of the labels in b-c between the caption and the figure. The text description refers to the caption: “inactivated MRs were significantly more conserved than activated MRs, both across models and lineages”. But as it is one of the major point of the authors to define subsequent analysis, the authors should provide the raw data of the similarity of VIPER-inferred activity analysis to verify that there is no mismatch in the interpretation.

Thanks for pointing out this error. We fixed the labels in Supplementary Fig. 2b and c. The VIPER-inferred differential protein activity for each model and time point is now provided by Supplementary Table 2.

8. The authors suggest the binding of viral proteins to inactivated MRs indicates a host immunity evasion mechanism by the virus. The authors should provide as supplementary data an overview of the identified viral proteins and their targets: is this a general pattern across viral proteins, which would be rather unexpected, or is the reported overlap due to overrepresentation of a few viral proteins? Moreover, the manuscript would benefit if the authors considered to reproduce the inactivated MR enrichment analysis on other interactome datasets.

This is an important point we wish to clarify. We do not suggest that binding of viral proteins to inactivated MRs indicates a host immunity evasion mechanism. We simply show that host proteins previously identified as direct binders of viral proteins by Gordon et.al (Nature 583:7816, 2020) are enriched among the host candidate MRs we empirically identified as the proteins most inactivated in response to viral infection. This set of inactivated host factors are *not* enriched in innate immune-related gene sets, but conversely in cell proliferation hallmarks (E2F, MYC and G2M checkpoint), DNA repair and PI3K/AKT/MTOR pathways (Manuscript Fig. 1b).

Reviewer #2

Specific points

1. The authors themselves are aware of the limitations of this approach, where the appropriate physiological cellular models are required to obtain gene signatures between infected and non-infected cells, and for drug selection from drug perturbation databases. The authors initially assess lung and GI-derived cells; however, based on the small selection of cell types from the PanACEA database, proceed to focus only on GI-lineage cells. To this end, the final treatments may be more focused in treating GI infected cells (as highlighted by the authors' subsequent testing on lung adenocarcinoma cells), rather than lung-infected cells, which is the primary site of infection. This results in the exclusion of a large number of lung-specific drugs, and is possibly a limitation of re-purposing an oncology database, and limits the overall usefulness of this study. Would the authors be able to find alternative drug perturbation profile databases that would be more appropriate for this study?

While the goal of this work is in part to present a novel framework for host-directed antiviral intervention—our main goal is to show the existence of the viral checkpoint and its requirement for effective virus replication (see response to reviewer #1, minor point #6)—, the pathogen and models evaluated were selected with the sole purpose of providing a proof-of-concept validation of the approach, and not to identify or repurpose specific drugs that can be directly translated to the clinic for COVID-19. Yet, as it turned out, some of the drugs identified by this approach are currently being evaluated in the clinical setting for COVID-19. We agree with the reviewer that the use of GI models would be a clear limitation for translating the particular drugs evaluated in this work to the clinic; and, moreover, a drug perturbational dataset obtained in adequate models representing lung context would be required. Unfortunately, the lung adenocarcinoma cell line that is part of the PanACEA drug perturbation repository does not constitute a good match, based on conservation of MR protein activity, to the viral checkpoint of the lung epithelial cells. We could not find any alternative dataset similar to PanACEA, providing gene expression profiles in response to equipotent, short-term drug perturbations in lung epithelial relevant models. In light of the importance of clarifying the translational relevance of our discoveries, we have clarified this limitation in the revised version of the manuscript (Please, see line 391).

2. The authors test their selected drugs on GI cells by pre-treating cells with the appropriate drug for 24 hours before SARS-CoV-2 infection. Although this prophylactic approach may be clinically useful in the future, current treatments for SARS-CoV-2 are therapeutic (ie given after initial infection), so it would be useful to know if similar results are seen if cells are treated with the desired drug after SARS-CoV-2 infection. It would also be interesting to know why the authors chose LoVo cell line to perform the initial drug screen, and Caco-2 to perform the infectivity assay.

This is an important point that we wish to clarify. The predicted effect of the drugs is to

lock the cell in a transcriptional state that is *refractory* to virus hijacking of the cell machinery for its replication. Based on this mechanistic basis, it's our view, in line with the reviewer's comment, that the drug could not rescue (i.e., prevent replication) in a cell whose regulatory, transcriptional, and metabolic machinery has been already hijacked by the virus. But at the "macro-host" level (i.e. patients), the natural history of SARS-CoV-2 infection undergoes multiple generations of infection cycles over time. Therefore, we can reasonably expect that a pharmacologic agent that is able to keep not-yet-infected cells in a state that is refractory to virus hijacking would effectively limit the availability of virus replication-susceptible host cells, thereby mitigating the spread and amplification of the infection as it evolves in the human host. In this regard, for example, interferons are potent antiviral cytokines. If cells are treated with Interferons prior to infection, they will show resistance to viral infection. However, if cells are treated with interferons just a few hours after infection, these cytokines do not induce, or only minimally induce, an antiviral state at the cellular level (Metz-zumaran et al., J. Virol , 2022, doi: 10.1128/jvi.01705-21).

The LoVo cell line was selected for the PANACEA resource based on its recapitulation of tumor MR protein activity (please, see <https://www.biorxiv.org/content/10.1101/2020.12.21.423514v2>), and it is the only colorectal model included in the PANACEA perturbational dataset. Fortunately, the activity of the MR proteins of the Caco-2 cell line, a colorectal carcinoma model susceptible to SARS-CoV-2 infection, is conserved in the LoVo cell line, as shown in Supplementary Fig. 5a-b, supporting LoVo as an adequate model to elucidate drug mechanism of action for the Caco-2 cell line model (please, see <https://www.biorxiv.org/content/10.1101/677435v1>).

3. The authors seed 20,000 cells/well at 0 hours and then treat with the appropriate drug for 24 hours before replacing with fresh media and infecting with SARS-CoV-2. During this period, would some cell death occur, especially at higher concentrations; therefore would there be the same number of cells across all concentrations at the time of infection? Hence the MOI would be inaccurate if there were different cell numbers. It would be perhaps better to pre-treat the cells in a flask for 24 hours, then re-seed for infection.

We thank the reviewer for this comment. The reviewer is correct. For some drugs, in high concentration conditions, cytotoxicity occurs and, accordingly, at 24 hrs post treatment, some cell death can be observed. In these conditions, using the same amount (volume) of virus will result in a higher MOI, making interpretation of the results difficult with an over-conservative tendency when evaluated the anti-viral effect of the tested drugs. To alleviate this problem, the viability-normalized effect on SARS-CoV-2 replication (antiviral effect) was quantified as the log-ratio between viral replication and cell viability reduction relative to vehicle-treated (DMSO) controls. This allows us to distinguish cytotoxic effects vs antiviral effects. For all drugs selected, the cytotoxicity

was present at higher concentrations than the antiviral effects determined at these concentrations (Supplementary Fig. 6a). Moreover, when we validate our approach using a non-related virus (rotavirus) using the same compounds on the same cells (same cytotoxicity), we do not see an inhibition of rotavirus infection, further arguing that we are measuring real antiviral activity.

4. The authors use immunofluorescence to measure viral infection (anti-dsRNA antibody) and cell viability (Draq5). However, in the methods, the authors fix and permeabilize the cells first (for anti-dsRNA staining), and then stain with Draq5; this would have the effect of staining all host cells, including both live and dead cells. The authors see a drop in “cell viability” at high concentrations which could justify dead cells at high concentrations not binding the Draq5 dye – however, truly dead cells may have rounded up and floated off into the media during initial 24 hour drug treatment, and after fixation and washing. Therefore, Draq5 may capture live and dead cells still stuck to the plate at lower concentrations, but properly dead cells would have been removed and therefore no fluorescence can be seen at higher concentrations. As the authors clearly state that it is important to find a drug treatment that stops viral replication but keeps cells alive, it would be useful if they could use a stain that specifically measures live cells only, for example Calcein AM which relies on cellular enzymatic activity in live cells for fluorescence, rather than the use of DNA intercalating dyes, which can stain both live and dead cells indiscriminately. This is of particular importance as the authors calculate the antiviral effect based on the log-ratio between viral replication and cell viability reduction.

We thank the reviewer for this well-reasoned comment. In agreement with the reviewer, we understand the potential problem inherent to our Draq5 cell viability method. We have performed a cytotoxicity evaluation using a more classical approach based on Lactate dehydrogenase (LDH) assay (Promega G1780) on the following drugs (Amioradone, Bicalutamide, Bosutinib, cyclosporine, Osimertinib, Vorinostat). As can be seen in the titration below (Figure 4), some compounds show toxicity at their highest concentration similar to what was observed in the Drag 5 method (Supplementary Fig. 6). Due to the consistency of both assays, we believe that in our setup, the Draq5 approach is reliable. Importantly, and as mentioned above in point 3, we used the same approach with rotavirus using the same cells and the same compounds that were found active on SARS-CoV-2 infection. Our results show that these compounds do not have an effect on rotavirus infection. If inhibition of SARS-CoV-2 infection was the result of cytotoxicity induced by the drugs, leaving dead cells attached to the coverslip which are then stained with Draq5, a similar inhibition of rotavirus infection should have been observed.

Figure 4. Effects of drugs on cell number vs cell viability. Caco-2 cells were seeded in 96-well plates. 24 hours post-seeding the indicated drugs were added to cells with increasing concentrations. Cell number was assayed 48 hours later by DraQ5 staining. Cell Viability was assayed 48h later by LDH assay (Promega CytoTox). All samples were performed in triplicate. Error bars indicate the standard deviation.

Minor points

1. The Introduction needs to be revised for clarity and content. There needs to be more detail on what master regulator proteins are for those not familiar with the terminology, and their importance (including relevant references). Certain paragraphs of the

Introduction would be more appropriate for the Results and Discussion ie where the authors describe and justify in detail their approach to this proof-of-concept study.

We agree with the reviewer and have added the following sentences to the first paragraph of the introduction: “MRs are proteins whose activity is necessary and sufficient to maintain the transcriptional identity of a specific cellular phenotype. They are organized in highly inter-regulated protein modules, or transcriptional regulatory checkpoints, which operate as a molecular switch, controlling the transcriptional identity of both physiologic and pathologic cell states (for a recent perspective, please see ⁷).” (see line 69).

2. Certain sections of the Results would benefit from more detail; for example, in the section titled “SARS-CoV-2 induced MR signature”, it is not mentioned that VIPER is the algorithm used to analyse single cell RNA sequencing data. Line 141 states: “Single cell analysis revealed highly conserved differential protein activity signatures, as defined by the top 50 most differentially active candidate MRs ...” – is this analysis conducted using VIPER? Also, it is stated two sections later that “5,734 (proteins) we analysed by VIPER” (line 187) – this is not mentioned in the first section. This could be clarified by moving line 562 in Methods (“The differential activity of 5,734 proteins, including 1,723 transcription factors, 630 co-transcription factors, and 3,381 signaling proteins, was estimated for each of the differential gene expression signatures with the VIPER algorithm ...”) to the appropriate Results paragraph. It would also be useful to conclude each Results section with an appropriate summarizing sentence, to help the reader understand and interpret the results stated.

We appreciate the useful comments which have improved the clarity of our manuscript. We followed the reviewer suggestion, indicating that the analysis of the single cells has been performed with the VIPER algorithm (See line 147). We also reviewed all results’ subsections and incorporated a closing statement to aid the reader’s interpretation of the results (see lines 162, 185, 201, 265, 293, 304, 317 and 328).

3. Line 142: is there a reason why there is a focus on only the top 50 candidate proteins? How was the number 50 chosen?

The use of 50 proteins is based on previous results in the context of cancer, where less than 50 candidate MRs are required to canalize the effect of each individual patient genetics on the cell transcriptional identity (Paull et.al. 2021, Cell 184:334). We have added this justification in the results section, line 151.

4. Please clarify if the findings in line 160 – 162 are from bulk or single cell sequencing as it is not stated here in the Results.

We agree with the reviewer that it is not clear whether these results are based on bulk or sc-RNA-Seq. These results are related to bulk sequenced samples, and we have clarified this in the text in line 171.

5. Line 218: please add more detail to explain the use and purpose of OncoMatch; this seemed vital in excluding lung lineage lines from further analysis, and so should be expanded upon.

Thanks for highlighting this point. We have added a sentence indicating the importance of using models recapitulating the MR protein activity to maximize the drug MOA conservation and sensitivity of the OncoTreat analysis, as we have previously observed. See line 244.

6. Line 237: please provide a short explanation of the aREA algorithm

We added a short explanation for aREA in line 262.

7. Figure 1a: legend states top 10 genes, but there appears to be more genes shown on heatmap?

The top most activated proteins in response to SARS-CoV-2 infection for each of the 10 evaluated conditions, including 4 different models and different time points for each model (62 proteins across all conditions), are shown in the left heatmap of Fig. 1a, and the top 10 most inactivated proteins in response to SARS-CoV-2 infection for each of the 10 evaluated conditions (69 proteins across all conditions) are shown in the right heatmap in Fig. 1a. We have clarified this point in the legend of Fig. 1.

8. Figure 1a: figure shows NES color bar but no explanation in legend; does darker colour indicate enrichment or expression/activity?

The heatmap colors indicate differential protein activity expressed in Normalized Enrichment Score (NES) units, with proteins inactivated and activated in response to SARS-CoV-2 infection shown in blue and red color, respectively. We have clarified this in Fig. 1 legend.

9. Figure 3: difficult to interpret, would benefit from adding an additional summary sentence in the relevant Results section to aid in understanding this figure, for example after the following sentence in lines 239 – 241: “Among the 154 FDA-approved drugs profiled in LoVo cells, ViroTreat prioritized 22 (13 orally available and 9 intravenous) at a

highly conservative statistical threshold ($p < 10^{-5}$, Bonferroni corrected (BC)), see Fig. 3 and Supplementary Table 2).” – therefore can it be concluded that these top 22 drugs are predicted to significantly activate more of the top 50 genes/proteins that are inactivated after SARS-CoV-2 infection?

We appreciate the reviewer’s comment and following their suggestion, we added a sentence summarizing the interpretation of these results at the point the reviewer suggested. See line 267.

10. Figure 4a: well reflections make figure hard to interpret.

We agree with the reviewer but since these are microplates photographs, the only way we see possible to remove such reflections being produced by the plastic well’s wall is by editing the images and manually removing those pixels. However, we prefer to show the original images (non-edited).

11. Figure 4b: is it possible that some drugs can show high infectivity reduction but no inhibition? Needs more detail on what is being inhibited (viral replication?)

In this biological model, we are currently quantifying dsRNA, which is the product of viral genome replication. Yet, it is not possible to dissect whether the effect of the drug is on viral genome replication per se or on infectivity (virus entry), since both would lead to a reduced amount of dsRNA. We included a sentence clarifying this point in line 281.

12. Figure 4c: What does the dotted line show?

The dotted line indicates the diagonal in the ROC plot, which is the expected result if drugs were prioritized uniformly at random.

13. As mentioned in point 2, certain sections of the Methods could be moved to the Results to aid the reader’s understanding of the paper. For example, lines 664 – 667 describing ViroTreat could be moved to the relevant Results paragraph.

We agree that moving these sentences to relevant parts of the results section will aid the readers to understand the manuscript. In this regard, we have moved the sentences highlighted by this reviewer to the results section. See line 219.

14. Methods: RNA isolation, cDNA and RT-qPCR: need to correct spelling of manufacturer’s

We thank the reviewer for finding this typo.

15. Supplementary Figure 1: figure legend would be more informative if provided with numbered sub-sections 1-6 to match figure workflow.

We have expanded the description of this figure in the legend.

16. Supplementary Figure 6: is it necessary to show multiple graphs for the same drug eg Amiodarone – if they are showing different concentrations, is it possible to merge the data onto a single graph?

We understand that the number of plots can be reduced by including the different experiments for the same drug in the same plot. However, we respectfully disagree with the reviewer's suggestion on this particular point. The separate plots represent independent experiments and the analysis of the data is performed using the experiment-matched controls. Thus, we think including all the experiments in the same plot will make it more difficult to interpret by the readers.

17. Supplementary Figure 9: what do the other colored lines show?

The colored lines represent the 4 different gaussian components of the mixed-gaussian model fitted to the data distribution. We have clarified this in the supplementary figure legend.

REVIEWERS' COMMENTS:

Reviewer #1 (Remarks to the Author):

The authors adequately addressed my points.

I support acceptance of the manuscript in Communications biology.